# The Role of Epigenetic Biomarkers as Diagnostic, Predictive and Prognostic Factors in Colorectal Cancer

**DOI:** 10.3390/cancers17162632

**Published:** 2025-08-12

**Authors:** Zuzanna Chilimoniuk, Konrad Gładysz, Natalia Moniczewska, Katarzyna Chawrylak, Zuzanna Pelc, Radosław Mlak

**Affiliations:** 1Department of Laboratory Diagnostics, Medical University of Lublin, 20-093 Lublin, Poland; 52705@umlub.edu.pl (K.G.); natalia.jaszek@umlub.pl (N.M.); 2Department of Surgical Oncology, Medical University of Lublin, Radziwiłłowska 13 St., 20-080 Lublin, Poland; 56646@student.umlub.pl (K.C.); zuzanna.pelc@umlub.pl (Z.P.)

**Keywords:** CRC diagnosis, epigenetic, biomarkers, CRC prediction, CRC prognosis, DNA methylation

## Abstract

Recent advances in cancer research emphasize the urgent need for novel, accurate, and minimally invasive diagnostic tools for colorectal cancer. Current screening methods often suffer from limited sensitivity, specificity, and can be costly or invasive. Epigenetic alterations, including gene methylation patterns and non-coding RNA expression, have emerged as promising biomarkers due to their detectability in patient tissues and fluids. This review critically evaluates the potential of these epigenetic factors in colorectal cancer diagnosis and prognosis. Despite promising findings, many identified markers lack sufficient diagnostic accuracy, underscoring the necessity for continued research. The identification of reliable epigenetic biomarkers could revolutionize early detection strategies, enabling better patient stratification, improved treatment personalization, and ultimately, enhanced clinical outcomes.

## 1. Introduction

Colorectal cancer (CRC) remains one of the most commonly diagnosed malignancies worldwide, with an estimated 1,926,118 new cases reported in 2022, ranking it third among all cancers in terms of incidence. Moreover, CRC is the second leading cause of cancer-related deaths globally, accounting for approximately 9.3% of all cancer mortalities. These statistics underscore its significant impact on public health [1]. According to estimates by the International Agency for Research on Cancer (IARC), the incidence of CRC could increase by 63% to 3.2 million new cases per year by 2040, while mortality will rise to 1.6 million per year. In addition, the increase in incidence was found to be more marked in developing countries and among young adults [2].

Regarding etiology, there are three main types of CRC: sporadic, hereditary and colitis-associated, while risk factors can be divided into modifiable and non-modifiable ones. The latter group primarily includes age, gender and genetic syndromes. The disease is most common in patients over 50 years of age and is more prevalent in men than in women [3]. Hereditary non-polyposis colorectal cancer (HNPCC) and familial adenomatous polyposis (FAP) are the two most common forms of hereditary CRC, with untreated FAP resulting in a nearly 100% chance of developing CRC. Inflammatory bowel diseases, especially ulcerative colitis, also significantly increase the risk [4,5]. A family history of CRC is the strongest known risk factor for the condition. An individual’s chance of developing CRC is two to four times higher than that of someone without a family history of the disease [6]. Nonetheless, the majority of CRC cases (about 50–60% in the United States) are linked to modifiable variables, such as obesity, drinking alcohol, nicotinism, high red meat consumption and insufficient dietary fiber, as well as low physical activity [7].

As many as 25% of CRC patients are diagnosed at an advanced stage of the disease, which is associated with a poor prognosis and a low five-year survival rate of just 14%. This translates into a predicted annual death toll of 550,000 globally [8]. Detection of the disease at an early stage has a positive impact on patient survival time and treatment outcomes. Currently available diagnostic methods can detect about 40% of early CRC. There is no doubt, therefore, that the drive to improve early diagnosis and the search for new diagnostic biomarkers is a justified effort [9].

CRC predominantly develops from precancerous polyps, categorized into traditional tubular adenomas and serrated polyps. These lesions originate from the dysregulation of DNA repair and cellular proliferation pathways [10]. Under normal conditions, epithelial turnover occurs through controlled proliferation at the crypt base, followed by differentiation and apoptosis as cells migrate towards the colonic lumen. Mutations in these processes disrupt cellular homeostasis, leading to adenoma formation. Over time, these polyps grow, exhibit progressive dysplasia, and may acquire invasive characteristics [11].

CRC oncogenesis is driven by the involvement of numerous genes and the interplay of various signaling pathways, though the precise mechanisms remain incompletely understood. A substantial number of CRC cases are sporadic, progressing chromosomal gradually through the adenoma-carcinoma sequence. Approximately 10% of adenomatous polyps progress to adenocarcinoma, with the risk increasing proportionally to polyp size [2]. We can distinguish four main mechanisms of genetic changes in CRC: microsatellite instability (MSI), chromosomal instability (CIN), CpG island methylator phenotype (CIMP), and b-Raf proto-oncogene, serine/threonine kinase (*BRAF*) or Kirsten rat sarcoma viral oncogene homolog (*KRAS*) mutations [12].

The CIN pathway is the most common, present in 65–70% of sporadic colorectal tumors and is marked by chromosomal changes, including somatic copy number alterations (SCNAs), typically resulting from defects in chromosomal segregation [13]. These abnormalities are accompanied by mutations in tumor suppressor genes such as adenomatous polyposis coli (*APC*) and tumor protein p53 (*TP53*), as well as activating mutations in *KRAS* and phosphatidylinositol-4,5-bisphosphate 3-kinase catalytic subunit alpha (*PIK3CA*). The inactivation of *APC* is considered the initial genetic event in colorectal tumorigenesis [14]. Loss of *APC* function leads to nuclear accumulation of β-catenin, activating the Wnt signaling pathway. This activation drives the expression of Wnt-regulated genes associated with tumorigenesis, including *MYC*, cyclin D1 (*CCND1*), vascular endothelial growth factor (*VEGF*), and peroxisome proliferator-activated receptor delta (*PPARδ*). Mutations in other Wnt pathway components, such as *AXIN1*, *AXIN2*, or catenin beta-1 (*CTNNB1*), can further enhance Wnt signaling even in the absence of *APC* mutations [13,14]. Activating mutations in *KRAS* typically occur after *APC* mutations. *KRAS* functions within growth factor signaling pathways, including the epidermal growth factor receptor (EGFR) pathway. Mutant *KRAS* leads to constitutive activation of downstream pathways, including the Raf-MEK-ERK cascade, PI3K signaling via mTOR, and NF-κB-mediated transcription [15]. In the Raf-MEK-ERK axis, Raf kinases activate MEK1/MEK2, which phosphorylate ERK1/ERK2, driving cell cycle progression through phosphorylation of key enzymes. This signaling pathway is frequently activated in various tumors, including CRC, and serves as a promising therapeutic target, especially in the treatment of metastatic colorectal malignancies [15,16].

In addition to genetic changes, epigenetic mechanisms play an important role in the pathogenesis of CRC. Epigenetics refers to heritable modifications in chromatin structure and biochemical properties that occur without alterations to the underlying DNA sequence [17]. These mechanisms influence a wide range of physiological and pathological processes by modulating gene expression through alterations in the local and global accessibility of the chromatin to epigenetic markers [18].

DNA methylation represents one of the most widespread epigenetic modifications governing gene expression. The most extensively studied form of this process involves the enzymatic addition of a methyl group (CH_3_) at the C5 position of the cytosine ring by DNA methyltransferases (DNMTs), resulting in the formation of 5-methylcytosine [19]. This modification predominantly occurs at CpG dinucleotides, regions where a cytosine nucleotide is immediately followed by a guanine nucleotide in the 5′ to 3′ orientation, linked via a C-phosphodiester-G bond [20]. The majority of CpG sites within the genome are methylated, including those located within gene bodies. In contrast, CpG islands—regions characterized by a high density of CpG sites—are typically unmethylated and are present in approximately 60–70% of gene promoters [21]. Compelling evidence indicates that hypermethylation of CpG islands within promoter regions is correlated with the transcriptional silencing of tumor-suppressor genes in cancer cells. In CRC such aberrant hypermethylation has been observed in the promoters of key tumor-suppressor genes, such as cyclin-dependent kinase inhibitor 2A (*CDKN2A*), MutL protein homolog 1 (*MLH1*), and *APC* [22]. Conversely, another epigenetic anomaly in CRC carcinogenesis is genome-wide hypomethylation, which often arises during the early stages of tumor development. This global DNA hypomethylation activates proto-oncogenes in promoter regions, potentially leading to the loss of genomic imprinting—such as in the case of *IGF2*—or directly driving the activation of oncogenes, including *MYC* and Harvey rat sarcoma virus (*HRAS*) [23,24].

The most well-studied post-translational histone modifications that alter gene expression in CRC carcinogenesis are acetylation and methylation. Acetylation status can have a dual effect. Hyperacetylation of histones associated with protooncogenes activates gene expression, while hypoacetylation of histones associated with tumor suppressor genes silences the corresponding genes [25]. Histone methylation not only modulates the compaction and structural state of chromatin but also generates specific docking sites that are recognized by various chromatin-associated proteins. This dynamic modification is mediated by the opposing activities of histone methyltransferases (HMTs), which catalyze the addition of methyl groups, and histone demethylases (HDMs), which remove them [26,27]. Dysregulation of these enzymes, either through overexpression or underexpression, can disrupt the global histone methylation landscape, leading to widespread alterations in the transcriptional regulation of numerous oncogenes and tumor-suppressor genes, thereby contributing to the progression of CRC [27].

Non-coding RNAs (ncRNAs) comprise a diverse class of RNA transcripts that do not encode proteins but are integral to numerous cellular processes, including gene activation and silencing, RNA splicing, modification, editing, and protein translation [28]. The primary subcategories of ncRNAs include microRNAs (miRNAs), long non-coding RNAs (lncRNAs), and circular RNAs (circRNAs). Collectively, ncRNAs account for approximately 76–97% of the human genome and serve as critical regulators of cellular homeostasis. Their aberrant expression, particularly in cancer, highlights their dual roles as potential oncogenes or tumor suppressors, making them promising targets for therapeutic intervention and early diagnostics [28,29].

MiRNAs are small, non-coding, single-stranded RNA molecules that modulate gene expression at the post-transcriptional level. They regulate the translation of over half of protein-coding genes, including those involved in critical cancer-related processes such as cell proliferation, differentiation, and apoptosis [30]. MiRNAs exert their regulatory functions either by targeting specific mRNAs or by acting as global modulators of gene expression. Aberrant miRNA expression can result from both DNA hypermethylation and hypomethylation, which influence their transcriptional activity [31]. Numerous studies have reported significant differences in miRNA expression profiles between CRC tissues and adjacent normal tissues. In cancerous tissues, miRNAs can be either upregulated or downregulated, although a greater proportion tend to be overexpressed rather than suppressed [30,32].

LncRNAs have been defined as non-coding transcripts over 200 nucleotides in length. They take part in a wide range of biological processes, including cell proliferation, differentiation, apoptosis and stem cell self-renewal [33]. Having such a wide range of functions, they influence virtually all steps in CRC carcinogenesis. They play a role in the WNT, EGFR, TGF-β and p53 signaling pathways, as well as influencing progression and metastasis [34,35].

CircRNAs are a distinct class of single-stranded RNA molecules with a covalently closed loop structure, lacking both 5′ and 3′ ends as well as poly(A) tails. This unique configuration renders them resistant to degradation by RNase R, making them significantly more stable than linear RNAs [36]. Functionally, circRNAs serve as key regulators of gene expression, participating in various biological processes such as acting as miRNA sponges, modulating transcription and translation, interacting with RNA-binding proteins (RBPs), and facilitating the translation of peptides and proteins. Emerging evidence specifically links circRNA abnormalities to the progression and development of CRC [37,38].

The epigenetic modifications described above do not operate in an isolated manner but rather constitute a complex regulatory network with significant interactions between them. An example that occurs during CRC carcinogenesis may be miRNAs, which themselves are subject to epigenetic regulation through methylation of their promoter regions [39]. According to the latest knowledge, the pathogenesis of cancer is influenced by the interplay of genetic and epigenetic processes. Genetic mutations in genes encoding epigenetic effectors result in their impaired activity and can alter gene expression, while genetic mutations in regulatory sequences can directly affect gene expression programs because they prevent the binding of transcription factors and epigenetic modifiers [40].

Most CRCs develop slowly from precancerous polyps, allowing for early detection and screening of both lesions and early-stage cancer [41]. Given the strong link between prognosis and early diagnosis, CRC screening is generally recommended from the age of fifty [42]. When detected early, the 5-year survival rate is approximately 90%, but this significantly drops once the cancer spreads beyond the colon or rectum [43]. Globally, most screening programs rely on fecal occult blood tests. Fecal Immunochemical Tests (FITs), which detect human hemoglobin using specific antibodies, offer improved specificity over guaiac-based tests. FITs require only a single home-collected sample and are unaffected by diet or medication, increasing patient compliance. They are also less expensive than invasive methods [44,45]. Colonoscopy typically follows a positive non-invasive test and allows full colon examination and removal of polyps, making it the most effective CRC prevention tool [46]. However, its downsides include high cost, need for trained specialists, preparation requirements, and patient reluctance due to its invasive nature, potential complications (e.g., bleeding, perforation), and the need for 10-year follow-up intervals [47].

Modern CRC treatment follows a multimodal approach, combining chemotherapy, radiotherapy, targeted molecular therapies, and surgery. Neoadjuvant chemotherapy aims to downstage tumors and control metastases to improve surgical outcomes, while adjuvant chemotherapy is used post-surgery to reduce recurrence risk and improve prognosis [48]. Personalized therapy—using monoclonal antibodies or small-molecule inhibitors tailored to tumor molecular profiles—is gaining ground [49]. For locally advanced rectal cancer, radiotherapy helps shrink tumors and preserve anal sphincter function [50]. Despite advancements, surgical resection remains the only curative option for advanced CRC. Current standards include complete mesocolic excision with central vascular ligation or D3 dissection, and total mesorectal excision for rectal cancer [51]. Treatment effectiveness depends largely on cancer stage at diagnosis and complications such as malnutrition, particularly cancer cachexia [52].

The tumor-node-metastasis (TNM) staging system remains the gold standard for prognostic evaluation in newly diagnosed CRC. Survival decreases with advancing tumor (T) stage—5-year overall survival (OS) rates are 87.5% for T3, 71.5% for T4, and just 46% for T4b tumors. Lymph node involvement (N stage) is the second most critical prognostic factor after distant metastasis, with node-positive patients showing 5-year OS of 30–60%, versus 70–90% in node-negative cases. The presence of distant metastases (M stage, stage IV disease) is the strongest negative prognostic indicator, with a 5-year OS below 10% [53]. Between 35% and 50% of CRC patients have metastases at diagnosis, with 20% presenting with them initially. Even among those diagnosed with localized disease, up to 50% may later develop metastases [54]. Early screening, typically recommended after age 50, is vital as prognosis is closely tied to disease stage: about 90% 5-year survival in stage I versus less than 10% in stage IV [43].

Given the limitations of traditional diagnostic methods, the development of new, cost-effective assays is essential for improving CRC detection. Epigenetic changes have emerged as promising biomarkers for early CRC diagnosis due to their ease of detection in body fluids and tissues. Although many potential biomarkers are rapidly being identified, numerous candidates still lack sufficient sensitivity and specificity. This underscores the ongoing need for research aimed at finding more reliable biomarkers to help reduce reliance on invasive diagnostic procedures. Our review will focus on assessing the potential of specific epigenetic factors as biomarkers in CRC. In all tables, the order of biomarkers cited refers to the order in which they are described in the text.

## 2. Diagnosis of CRC

### 2.1. Cohort Studies

To date, there have been many cohort studies on the use of epigenetic changes in CRC diagnosis. These studies involve the analysis of individual biomarkers, most often in stool or blood samples. However, despite surprisingly good sensitivity and specificity results for CRC diagnosis, many of them still require further research and clinical validation. Several non-invasive screening tests for CRC detection are currently available on the market. So far, the FDA has approved two tests based on epigenetic biomarkers [55,56,57]. One of them is a test called Epi proColon 2.0, which is based on the detection of changes in Septin 9 (*SEPT9*) methylation in the plasma of patients undergoing testing. The level of *SEPT9* methylation increases with the severity of the disease [58]. The Epi proColon 2.0 test was approved in 2016 by the FDA for screening for CRC in adults over 50 years of age. Unfortunately, this test has a relatively low sensitivity for detecting advanced adenoma (AA), with values ranging from 75 to 79.3% for CRC detection and 27% for AA detection, with a specificity of 96.7–98% for CRC detection [57,59,60]. Another test approved by the FDA is called Cologuard (Exact Sciences). It is also the first multitarget, stool-based DNA test to be approved for screening and diagnostic use in people over 45 years of age [57]. This test is based on a combined analysis of methylation of the N-Myc Downstream Regulated Gene (*NDRG*) and Bone morphogenetic protein 3 (*BMP3*) genes, 7 mutations in the *KRAS* gene, and an immunochemical test for human hemoglobin. The reference gene for the amount of human DNA is *β-actin*. The sensitivity of this test ranges from 42 to 46.2% for AA detection to 92.3% for CRC detection, while the specificity is approximately 89% [57,61,62]. Despite FD approval, there is still a need to search for highly sensitive and specific biomarkers for CRC detection, especially considering the low diagnostic efficacy of these tests in detecting early stages of the disease and AA.

#### 2.1.1. DNA Methylation

##### Stool

Over the past 5 years, numerous studies have been published on new possibilities for using changes in gene methylation status as biomarkers for CRC. In 2020, Liu et al. analyzed 324 methylated genes in CRC cell lines, from which 10 hypermethylated genes were selected for analysis in tumor tissue samples. Ultimately, the four most promising biomarkers were tested for their diagnostic potential in CRC in patient stool samples. All four genes were found to have very high sensitivity and specificity values, which were 92.5% and 91.6% (area under curve (AUC) = 0.97) for Collagen type IV alpha 2 chain (*COL4A2*), 88.8% and 88% (AUC = 0.97) for Collagen type IV alpha 1 chain (*COL4A1*), 88.8% and 96.4% (AUC = 0.96) for T-cell leukemia homeobox 2 (*TLX2*), and 82.5% and 96.4% (AUC = 0.95) for Integrin Subunit Alpha 4 (*ITGA4*) in distinguishing stool samples from CRC patients from healthy individuals. Additionally, the two biomarkers with the best results were combined into a panel consisting of *COL4A2* and *TLX2*, and when used together, the epigenetic changes gave a sensitivity and specificity of 91.3% and 97.6% (AUC = 0.98), respectively [63]. It was also shown that a methylation panel of three genes, Syndecan 2 (*SDC2*), *SEPT9* and Vimentin (*VIM*), achieved high diagnostic potential as biomarkers for diagnosing CRC in stool samples. The sensitivity and specificity of the panel were 91.4% and 100%, respectively (AUC = 0.99) [64]. Shen et al. discovered two new potential CRC methylation biomarkers (cg13096260 and cg12993163). In the validation group, their sensitivity and specificity were determined to be 92.6% and 90% (AUC = 0.94) for total CRC and 93.6% and 90% (AUC = 0.97) for early-stage CRC (stages I–II), respectively. These biomarkers were selected using bioinformatic tools and identified using methylation-specific quantitative PCR [65]. In 2020, Zhao et al. evaluated the diagnostic utility of methylation analysis of two genes in the context of early CRC detection using stool samples. The sensitivity of CRC detection using these two epigenetic changes was estimated to be 92.3% and the specificity was 93.2% (AUC = 0.98) in the validation group [66]. Surprisingly positive results were presented in the same year by Liu et al., who showed that the use of a combination of three DNA methylation biomarkers detected in stool, *SEPT9*, *SDC2* and Secreted Frizzled Related Protein 2 (*SFRP2*) showed sensitivity and specificity of 94.1% and 89.2% (AUC = 0.94), respectively, for the detection of CRC in a group of 1142 patients with intestinal abnormalities. This study used an alternative method of DNA methylation status analysis without hydrogen sulfide conversion, which was based on enzymatic digestion in CRC screening [67]. Wang et al. evaluated the diagnostic utility of a stool DNA test detecting the methylated *SDC2* gene in a large group of patients from three different centers. The sensitivity of the test showed very positive results, reaching 83.8% for CRC in general (AUC = 0.95 in terms of CRC detection) and 87% for early-stage CRC (I–II). The specificity of the test was 98% [68]. Promising results of the search for innovative epigenetic biomarkers among changes in DNA methylation status were published in 2024. Yun et al. developed a panel of 3 genes, PR domain containing 12 (*PRDM12*), Forkhead box E1 (*FOXE1*), and *SDC2*, whose methylation levels were analyzed in stool samples. The sensitivity and specificity of this panel were as high as 92.8% and 97.2% (AUC = 0.95), respectively [69]. Han et al. evaluated the possibility of using *SDC2* methylation level assessment in stool samples for the early diagnosis of CRC. The results were interesting, as the study showed sensitivity and specificity of 90.2% and 90.2% (AUC = 0.90) for CRC regardless of the stage of the disease [70]. The diagnostic potential of Potassium Voltage-gated Channel subfamily Q member 5 (*KCNQ5*) and Chromosome 9 open reading frame 50 (*C9orf50*) gene methylation in stool samples was also analyzed. It was shown that the sensitivity and specificity were 77.3% and 91.5% (AUC = 0.85) for *KCNQ5* and 85.9% and 95% (AUC = 0.90) for *C9orf50* [71]. A new study from 2024 presented promising results regarding CRC biomarkers: methylated *NDRG4* and *SDC2* genes. When the detection of methylation of both genes was combined, sensitivity and specificity reached 85.5% and 84.6% (AUC = 0.85) [72]. Pakbaz et al. evaluated the sensitivity and specificity of the *VIM* gene for CRC detection in stool. They were 60% and 100%, respectively [73]. Additionally, the diagnostic potential of a multi-target DNA methylation test consisting of the analysis of the methylation status of three genes, *SDC2*, Alcohol Dehydrogenase Iron Containing 1 (*ADHFE1*) and Protein Phosphatase 2, B’ gamma isoform (*PPP2R5C*), in stool samples was investigated. It was demonstrated that the sensitivity and specificity of such a panel are surprisingly high, reaching values of 91.5% and 90.3%, respectively [74]. In another study, a high diagnostic potential was reported for a panel consisting of *SDC2* and *SFRP2* methylation, *KRAS* gene mutation and the presence of human hemoglobin. The sensitivity and specificity of the panel reached 91.4% and 86.1%, respectively, for CRC detection [75]. Very interesting results were published by Zhang et al. in 2021, where they evaluated the diagnostic potential of *SDC2* and Tissue Factor Pathway Inhibitor 2 (*TFPI2*) gene methylation analysis in stool samples as a biomarker for early detection of CRC. The study showed that the use of a combination of two genes detected CRC with a sensitivity of 93.4% and a specificity of 94.3% [76].

##### Blood

A 2023 study by Dai et al. proposes a panel of six genes subject to methylation in CRC for use in cancer detection. The panel in the validation cohort showed sensitivity and specificity of 91.7% and 86.7% (AUC = 0.96), respectively, in the context of early CRC detection [77]. Analysis of the methylation status of the *TFPI2* and *NDRG4* genes in peripheral blood mononuclear cells has demonstrated high diagnostic potential, as both genes are hypermethylated in colorectal cancer. For the *TFPI2*, the sensitivity and specificity rates were 88% and 92% (AUC = 0.94), respectively, while for *NDRG4* they were 86% and 92% (AUC = 0.95), respectively [78]. Interestingly, Luo et al. analyzed the potential of using ctDNA methylation in the diagnosis of CRC. They found that a single methylation marker at the CpG site cg10673833 showed high diagnostic efficacy, as the sensitivity and specificity of this biomarker in plasma samples were 89.7% and 86.8% (AUC = 0.90), respectively [79]. In contrast, Zhang et al. showed the high diagnostic potential of their proposed biomarkers for detecting CRC. It turned out that the methylation panel of 4 genes, *C9orf50*, Potassium Inwardly Rectifying Channel subfamily J member 12 (*KCNJ12*), Zinc Finger Protein 132 (*ZNF132*) and twist related protein 1 (*TWIST1*), showed an overall sensitivity and specificity of 80% and 97.1% (AUC = 0.91), respectively [80]. A 2021 study on the hypermethylated myosin 1G (*MYO1-G*) gene showed that analysis of its methylation status in blood samples resulted in sensitivity and specificity of 84.3% and 94.5% (AUC = 0.94), respectively, in distinguishing CRC samples from healthy volunteers [81]. The combination of Polypeptide N-Acetylgalactosaminyltransferase 9 (*GALNT9*) and up-frameshift regulator of nonsense transcripts homolog A (*UPF3A*) gene methylation status developed by Gallardo-Gomez et al. presents results suggesting its potential use as a highly specific, non-invasive test for screening and early detection of CRC. This panel, analyzed in patient serum samples, achieved a sensitivity and specificity of 78.8% and 100% (AUC = 0.90), respectively, in distinguishing between patients with AA and CRC and healthy volunteers [82]. A study published by Shavali et al. examined the expression level and diagnostic potential of Heart and Neural Crest Derivatives Expressed 1 (*HAND1*) methylation. It was found that 83% of the patients studied showed hypermethylation of this gene. In addition, the level of CpG *HAND1* methylation showed sensitivity and specificity of approximately 93.3% and 80% (AUC = 0.85), respectively [83]. A study from 2023 by Lima et al. showed that in the Brazilian population, the combination of *SEPT9* and *BMP3* methylation status analysis in patient plasma showed sensitivity and specificity of 80% and 81%, respectively (AUC = 0.85). The authors therefore suggest the potential of these biomarkers for early diagnosis and screening for CRC [84]. In 2019, Li et al. investigated the methylation status of the *SFRP2* gene and its potential use as a single biomarker for CRC. Li et al. demonstrated that this gene is hypermethylated in most patients with CRC. The *SFRP2* MethylLight test was used to assess diagnostic utility, and patient serum was analyzed. The sensitivity and specificity of this test were 69.4% and 87.3% (AUC = 0.82), respectively [85]. Another proposal for a panel of biomarkers detected in blood was presented by Young et al. in 2021. It was found that a panel of methylated Interferon Regulatory Factor 4 (*IRF4*), IKAROS family Zinc Finger 1 (*IKZF1*) and Branched-chain amino acid transaminase 1 (*BCAT1*) genes effectively distinguished CRC patients from non-cancer patients, with a sensitivity and specificity of 73.9% and 90.1% (AUC = 0.82), respectively [86]. In 2024, the discovery of new genes with different methylation patterns in CRC was also reported, which could be used for non-invasive detection of CRC. Long et al. analyzed peripheral blood samples in which the diagnostic potential of methylation of three genes, Disabled homolog 1 (*DAB1*), *PPP2R5C* and Family with sequence similarity 19 member A5 (*FAM19A5*), was investigated based on cfDNA. It turned out that the combination of these three biomarkers resulted in a sensitivity and specificity for CRC detection of 64.2% and 78.4% (AUC = 0.76), respectively [87]. A recent study presented a new panel of 6 DNA methylation markers with a sensitivity and specificity for CRC detection of 67.5% and 98.2%, respectively. Their diagnostic potential was assessed in plasma samples. The methylation panel reported by the authors consists of genes such as *R16*, *F9*, *F8*, *R13*, *QKI* and *NDRG4* [88]. Jensen et al. analyzed the methylation panel of three genes: *C9orf50*, *KCNQ5* and CAP-Gly domain containing linker protein family member 4 (*CLIP4*) in plasma samples. This panel showed an average sensitivity of 85% and specificity of 99% in distinguishing CRC patients from healthy volunteers [89]. Another proposal was presented by Dastafkan et al., who assessed the methylation level of the promoter region of the Forkhead box protein F1 (*FOXF1*) gene. The results suggest that the promoter region of this gene is significantly more frequently methylated in patients with CRC compared to the control group. In addition, the potential of using this epigenetic change for early diagnosis of CRC was assessed, with sensitivity and specificity of 78% and 89.5%, respectively [90]. A meta-analysis conducted in 2022 demonstrated the high diagnostic utility of *SDC2* gene methylation for CRC detection. It showed that 11 different studies confirmed the possibility of using this epigenetic modification as a diagnostic biomarker for CRC with a combined sensitivity and specificity of 80% and 93%, respectively [91].

Currently, there are many scientific reports in the literature on the possibility of using the methylation of various genes for both the early diagnosis of CRC and screening for CRC. Despite many promising results and enormous diagnostic potential, many of these studies require further clinical validation. Some of them were conducted on too small a study group and require further validation studies involving many patients in a multicenter format to properly and objectively assess their potential for use in everyday clinical practice. Nevertheless, the analysis of methylation of selected genes certainly offers the possibility of replacing invasive or low-sensitivity and low-specificity methods of early diagnosis of CRC and screening in the future, which could result in a significant increase in patient survival rates.

A summary of the discussed DNA methylation-based biomarkers is provided in Table 1.

#### 2.1.2. ncRNA

##### miRNA

A recent scientific report presented an evaluation of the expression profiles and diagnostic relevance of five circulating miRNAs in the blood serum of CRC patients compared to healthy controls (HC). It was demonstrated that the expression level of miR-129 was significantly lower in CRC patients compared to HC. In addition, the expression levels of miR-410, miR-211, miR-139 and miR-197 were significantly higher in CRC patients compared to HC. The potential of these molecules as possible new diagnostic biomarkers for CRC was also analyzed. The sensitivity and specificity values were 100% and 100% (AUC = 1) (miR-211), 100% and 100% (AUC = 1) (miR-197), 70% and 60% (AUC = 0.73) (miR-139), 80% and 60% (AUC = 0.72) (miR-410) and 73% and 73% (AUC = 0.73) (miR-129) [92]. In contrast, Sabry et al. analyzed the diagnostic potential of miR-21 and miR-210 molecules and showed that when tested in serum, they exhibit sensitivity and specificity of 91.4% and 95% (AUC = 0.93) and 88.6% and 90.1% (AUC = 0.97), respectively. Both molecules were upregulated in CRC, which correlated with the stage of CRC [93]. In a study published by Herreros-Villanueva et al., a panel of 6 miRNA molecules tested in patient plasma was identified as a potential panel of epigenetic biomarkers for CRC. The sensitivity and specificity values of this panel were 85% and 90% (AUC = 0.92), respectively, suggesting great potential for use in the early detection of CRC [94]. A panel consisting of three miRNA molecules (miR-144-3p, miR-425-5p and miR-1260b) reported in 2019 by Tan et al. achieved very positive results in distinguishing plasma samples from CRC patients from healthy volunteers. The diagnostic potential of the proposed model is high, as evidenced by its sensitivity and specificity of 93.8% and 91.3% (AUC = 0.95), respectively [95]. In the search for novel non-invasive methods for CRC diagnosis, a potentially useful miRNA panel was identified. When analyzed in serum samples, it distinguished colorectal cancer patients from healthy individuals with a sensitivity and specificity of 91.8% and 91.7%, respectively (AUC = 0.95) [96]. In 2020, elevated serum levels of miR-21 were identified in CRC patients relative to HC. With a sensitivity of 95.8% and specificity of 91.7% (AUC = 0.94), miR-21 demonstrates strong potential as a non-invasive biomarker for early CRC detection [97]. Interestingly, upregulation of miR-92a-1 in CRC patients suggests its potential as a diagnostic biomarker. When analyzed in blood serum, it distinguished CRC cases from HC with a sensitivity of 81.8% and specificity of 95.6% (AUC = 0.91) [98]. Analysis of the Egyptian population in 2020 revealed increased expression of miR-1246 and decreased expression of miR-451 in individuals diagnosed with colorectal cancer. In addition, the researchers investigated the potential of these molecules as non-invasive biomarkers detectable in blood serum. MiR-1246 showed sensitivity and specificity of 100% and 80% (AUC = 0.92), respectively, while these values for miR-451 were 73% and 80% (AUC = 0.76), respectively, in the context of CRC detection [99]. In 2020, Elaguiza et al. published data on the diagnostic potential of three molecules detected in serum: miR-18a, miR-21, and miR-92a. It was shown that in distinguishing CRC patients from healthy individuals, these molecules exhibited sensitivity and specificity of 84% and 84% (AUC = 0.91) for miR-18a, 84% and 90% (AUC = 0.92) for miR-21, and 66% and 68% (AUC = 0.67) for miR-92a. In contrast, the panel comprising all three potential biomarkers was characterized by sensitivity and specificity of 86% and 90% [100]. Wang et al. demonstrated that miR-378e achieved sensitivity and specificity in the diagnosis of CRC of 89% and 80% (AUC = 0.93), respectively. Additionally, in combination with another biomarker, LI-cadherin, sensitivity and specificity increased to 86% and 94%, respectively [101]. A study from 2021 demonstrated that four miRNA molecules analyzed in serum successfully distinguished patients with TNM stage I colorectal cancer from healthy individuals. The sensitivity and specificity were 86.6% and 77.1% (AUC = 0.94) for miR-126, 83.3% and 85.7% (AUC = 0.92) for miR-1290, 89.9% and 74.3% (AUC = 0.89) for miR-23a and 90% and 71.4% (AUC = 0.88) for miR-940 [102]. Two other miRNA molecules studied in 2022 by Zhao et al. achieved sensitivity and specificity values of 87% and 100% (AUC = 0.97) for miR-627-5p and 93% and 70% (AUC = 0.90) for miR-199a-5p in distinguishing CRC patients from healthy individuals [103]. A study published in 2022 discovered two new biomarkers among miRNA molecules with very high sensitivity and specificity values. The authors, Mahmoud et al., suggest their possible use as new non-invasive biomarkers for the early detection of CRC. The level of expression differed significantly between the study groups of patients. The miR-223 molecule showed sensitivity and specificity of 97.1% and 96.7% (AUC = 0.96), while the miR-182 molecule showed 98% and 96% (AUC = 0.95), respectively [104]. In 2023, Pardini et al. published a paper presenting a panel of 5 miRNAs detected in patient stool samples. These included molecules such as miR-149-3p, miR-607-5p, miR-1246, miR-4488, and miR-6777-5p. This panel demonstrated a sensitivity and specificity of 82% and 91% (AUC = 0.95) for stage I–II CRC versus HC, and 90% and 88% (AUC = 0.96) for all CRC stages versus HC in the validation cohort [105]. Other promising diagnostic biomarkers were presented by Liu et al. in 2019. They considered two molecules: miR-1290 and miR-320d. It turned out that the free-circulating miRNA molecules analyzed in plasma were able to distinguish CRC patients from healthy individuals with a sensitivity and specificity of 76.7% and 90.2% (AUC = 0.88) for miR-1290 and 88.8% and 71.7% (AUC = 0.81) for miR-320d [106]. A diagnostic panel consisting of 4 miRNA molecules (miR-203a-3p, miR-145-5p, miR-375-3p, and miR-200c-3p) was proposed by Huang et al. in 2020 as a new non-invasive method for diagnosing and screening for CRC. The expression levels of these molecules analyzed in participants’ serum samples achieved a sensitivity and specificity of 81.3% and 73.3% (AUC = 0.89) [107]. Relatively good results were also presented by Peng et al., where a panel of three miRNA molecules (miR-30e-3p, miR-146a-5p, and miR-148a-3p) achieved sensitivity and specificity of 80% and 78.7% (AUC = 0.88), respectively, in distinguishing serum samples from CRC patients and healthy volunteers [108]. A paper published in 2020 by Pan and Miao demonstrated the high diagnostic potential of the miR-592 molecule detected in serum. The sensitivity and specificity in distinguishing CRC patients from healthy individuals were 82.8% and 78% (AUC = 0.84), respectively. In addition, these parameters were also satisfactory in distinguishing patients with early-stage CRC (stage I–II) from healthy individuals, at 78.6% and 80%, respectively (AUC = 0.80) [109]. In contrast, a 2020 study reported significantly elevated expression levels of miR-135b-5p in stool samples. The analysis of this molecule demonstrated notable diagnostic potential in distinguishing patients with advanced colorectal cancer (stage III/IV) from HC, achieving a sensitivity of 96.5% and specificity of 74.1% (AUC = 0.87) [110]. In 2022, Nakamura et al. presented a new proposal for a diagnostic panel consisting of four miRNA molecules (miR-193a-5p, miR-210, miR-513a-5p, and miR-628-3p), which effectively distinguished EOCRC (early-onset CRC) patients from healthy volunteers, with a sensitivity and specificity of 82% and 86% (AUC = 0.88), respectively, in the validation cohort [111]. Conversely, a 2021 study by Radwan et al., which also investigated the diagnostic potential of three miRNA molecules, demonstrated that the expression levels of these molecules in patient plasma yielded sensitivity and specificity values of 71% and 67% (AUC = 0.77) for miR-92a, 71% and 92% (AUC = 0.79) for miR-211, and 75% and 85% (AUC = 0.81) for miR-25, respectively [112]. Matboli et al. reported that the hsa-miR-3940-5p molecule is downregulated in patients with CRC. Furthermore, the diagnostic potential of this biomarker was assessed in blood serum samples, presenting a sensitivity of 93.5% and a specificity of 82.4% for CRC detection [113]. In 2021, Han et al. developed a diagnostic model consisting of three miRNA molecules: miR-15b, miR-21, and miR-31, achieving a sensitivity and specificity of 95.1% and 94.4% in distinguishing CRC from healthy cases [114]. Liu et al. published a meta-analysis on the diagnostic potential of miR-21, known as an oncogenic miRNA. The study showed that the sensitivity and specificity of this molecule for CRC, calculated on the basis of 18 different research studies, was 77% and 83%. The studies focused on the detection of circulating miR-21 molecules in the blood of patients and suggest their potential diagnostic usefulness [115].

##### lncRNA

Changes in the expression of ncRNA molecules are becoming increasingly popular as epigenetic biomarkers. In a previously mentioned study, the lncRNA SNHG14 molecule was found to be upregulated in patients with CRC. The diagnostic potential of this biomarker was further evaluated in blood serum samples, demonstrating a sensitivity of 98.5% and specificity of 90.8% (AUC = 0.95), indicating its high potential for CRC detection [113]. Another molecule investigated in this context is LINC02418. In 2019, it was reported that LINC02418 is significantly upregulated in individuals with CRC. Moreover, its diagnostic potential was evaluated, with exosomal LINC02418 analyzed in blood serum demonstrating a sensitivity of 95.2% and specificity of 66.4% (AUC = 0.90) in distinguishing CRC patients from the control group [116]. In contrast, Radwan et al. reported on the high diagnostic potential of lncRNA MALAT and PVT1 molecules, which were studied in the blood serum of patients. Both molecules were significantly upregulated in CRC. In terms of diagnostic utility, they achieved sensitivity and specificity values of 82% and 88% (AUC = 0.91) for MALAT1 and 90% and 70% (AUC = 0.85) for PVT1, respectively [117]. In 2022, Elabd et al. conducted research on the diagnostic potential of two lncRNA molecules, whose expression levels were analyzed in plasma samples. The LncRNA molecule ASB16-ASI showed sensitivity and specificity of 91.5% and 88% (AUC = 0.93), respectively, in distinguishing CRC patients from controls, and 88.2% and 62% (AUC = 0.95) in distinguishing early-stage CRC patients from controls. In contrast, the LncRNA AFAPI-ASI molecule showed sensitivity and specificity of 87.2% and 84% (AUC = 0.92) in distinguishing CRC from controls, and 70.6% and 84% (AUC = 0.93) in distinguishing early-stage CRC from controls [118]. In 2022, it was demonstrated that LINC01485 is significantly upregulated in patients with colorectal cancer. When detected in whole blood, this molecule exhibited a sensitivity of 98.3% and a specificity of 84% (AUC = 0.96) for distinguishing CRC patients from healthy individuals [119]. Other promising results were reported in 2023 concerning the diagnostic potential of lncRNA ZFAS and miR-200b in distinguishing colorectal cancer patients from HC. Both molecules, detected in serum, demonstrated high diagnostic accuracy, with miR-200b showing a sensitivity of 98.3% and specificity of 96.4% (AUC = 0.95), and ZFAS exhibiting a sensitivity of 92.9% (AUC = 0.95) [120]. In 2019, Liu et al. published data on a potential biomarker panel consisting of three lncRNA molecules upregulated in CRC: 91H, PVT-1, and MEG3. The sensitivity and specificity of this panel, tested in plasma samples to distinguish CRC patients from healthy volunteers, were 82.8% and 78.6% (AUC = 0.88), respectively [121]. Another interesting molecule is the lncRNA ADAMTS9-AS1, reported in 2019, whose expression level is significantly reduced in colorectal cancer. Its diagnostic potential was evaluated, with sensitivity and specificity in serum samples reaching 71.7% and 91.7%, respectively (AUC = 0.83) in the validation cohort [122]. In 2019, Jiang et al. evaluated the lncRNA NKILA molecule as a new potential diagnostic biomarker for CRC. Its expression level in CRC patients was found to be significantly reduced compared to healthy individuals. In addition, the sensitivity and specificity for early detection of CRC were 82.9% and 72.9% (AUC = 0.84), respectively [123]. A diagnostic panel developed in 2020 by Gharib et al., based on the expression analysis of 10 lncRNA molecules detected in stool samples, achieved a sensitivity of 74.9% and specificity of 94.1% (AUC = 0.85) in distinguishing patients across all stages of colorectal cancer from healthy individuals in the validation study. Additionally, the panel demonstrated a sensitivity of 68.2% and specificity of 83.7% (AUC = 0.82) in differentiating healthy individuals from patients with early-stage cancer (TNM stages I–II) within the same cohort. This panel consisted of 7 upregulated molecules (CCAT1, CCAT2, H19, HOTAIR, HULC, MALAT1, PCAT1) and 3 downregulated molecules (MEG3, PTENP1, TUSC7) in patients with CRC [124]. A 2022 study by Shen et al. demonstrated the diagnostic potential of the LINC01836 molecule, which is upregulated in CRC. Expression levels measured in blood serum samples showed a sensitivity of 65% and specificity of 87% (AUC = 0.81) [125]. In 2024, Li et al. discovered that the expression levels of three lncRNA molecules (lnc-PDZD8-1:5; NEAT1:11; LINC00910:16) were significantly elevated in individuals with CRC. In addition, the proposed diagnostic panel consisting of these molecules showed a sensitivity and specificity of 74.5% and 80.5% (AUC = 0.85) in the context of diagnosing individuals with CRC [126]. In a study published in 2024 by Lotfi et al., the expression levels and diagnostic potential of five lncRNA molecules in blood plasma were investigated. The researchers demonstrated that the expression levels of lncRNA CCAT1, BBOX1-AS1, and LINC00698 were significantly lower in patients with CRC compared to HC. In addition, the expression levels of FEZF1-AS1 and UICLM were significantly higher in patients with CRC compared to HC. The potential of these molecules as possible new diagnostic biomarkers for CRC was also analyzed. The sensitivity and specificity values were 85% and 93% (AUC = 0.88) (UICLM), 68% and 67% (AUC = 0.72) (BBOX1-AS1), 69% and 63% (AUC = 0.67) (FEZF1-AS1), 67% and 62% (AUC = 0.73) (CCAT1), and 67% and 60% (AUC = 0.70) (LINC00698) [92]. In 2020, the lncRNA DANCR molecule was also studied, and its expression level was analyzed in blood serum. It was found that the expression level was significantly elevated in samples from CRC patients, while the sensitivity and specificity in distinguishing CRC patients from healthy individuals were 67.5% and 82.5% (AUC = 0.75), respectively [127]. In contrast, a 2019 study by Wang et al. analyzed MEG3 expression levels in blood serum, reporting a sensitivity of 66.7% and specificity of 87.5% (AUC = 0.79) for colorectal cancer detection [128]. Another 2019 study presented the lncRNA-ATB molecule as a potential non-invasive biomarker for CRC detected in plasma. Abedini et al. showed that the sensitivity and specificity of the molecule were 82% and 75% (AUC = 0.78) in distinguishing between samples from CRC patients and healthy individuals. In addition, it was shown that the expression level of this lncRNA was significantly increased in CRC patients compared to the control group [129].

##### circRNA

CircRNAs, a type of ncRNA, have recently attracted considerable attention as potential biomarkers for CRC. In 2020, Xie et al. reported on the diagnostic potential of the circ-PNN molecule, which demonstrated sensitivity and specificity values of 89.7% and 69% (AUC = 0.83), respectively, in a validation study involving blood serum samples from 58 patients and 58 HC [130]. A 2019 study evaluated a panel of three circRNAs including circ CCDC66, circ ABCC1 and circ STIL, analyzed in plasma, which were significantly downregulated in CRC patients. This panel achieved sensitivity and specificity of 64.4% and 85.2% (AUC = 0.78), respectively [131]. Additionally, Zheng et al. investigated circLPAR1 combined with established biomarkers CEA and CA19-9 in plasma, resulting in a diagnostic panel sensitivity and specificity of 87.3% and 76.3% (AUC = 0.88) for CRC detection [132].

A summary of the discussed ncRNA-based biomarkers is provided in Table 2.

## 3. Prediction of Clinical Outcome in CRC

### 3.1. Epigenetic Factors in Prediction of Treatment Response

Although research on the potential of epigenetic factors as predictive biomarkers in CRC has grown in recent years, such studies remain relatively limited when compared to the larger body of work focused on their role in early detection or prognosis.

#### 3.1.1. DNA Methylation

A 2021 study by Jiang H. et al. highlighted the potential of DNA methylation as predictive biomarkers in CRC. They evaluated methylated septin 9 (*mSEPT9*) levels in 26 CRC patients undergoing chemotherapy (XELOX, FOLFOX, FOLFIRI), finding that 61.5% showed a decrease post-treatment, while 38.5% had increased levels. A reduction in *mSEPT9* correlated with improved treatment response, with 77.8% of these patients exhibiting non-progression of disease (Non-PD), compared to 22.2% with disease progression. *MSEPT9* was more accurate than CEA in predicting treatment response (69.2% vs. 53.8%) [133]. Boughanem H. et al. focused on long interspersed nuclear element-1 (*LINE-1*) methylation in CRC patients (*n* = 67). In this study the use of neoadjuvant therapy was significantly associated with higher *LINE-1* methylation (OR: 1.91; 95% CI: 1.43–2.56). In patients who received neoadjuvant therapy, significantly poorer recurrence-free survival (RFS) and OS were observed in those with low *LINE-1* methylation compared to those with high methylation levels [134]. In another study, Jamai D et al. examined Excision Repair 1, Endonuclease Non-Catalytic Subunit (*ERCC1*) and O6-methylguanine-DNA-methyltransferase (*MGMT*) methylation as a possible marker for predicting response to FOLFOX chemotherapy in CRC patients (*n* = 111). Both *ERCC1* and *MGMT* methylation exhibited a significant, negative correlation with resistance to FOLFOX chemotherapy (r = −0.575; r = −0.420, respectively), methylation level was significantly elevated in patients who responded to treatment compared to non-responders. It was also confirmed that simultaneous methylation of these repair genes was significantly, positively associated with perineural invasion (r = 0.417), distant metastasis (r = 0.605), cancer recurrence (r = 0.446) and resistance (r = 0.655) [135]. Hagiwara et al. examined whether promoter methylation of Checkpoint With Forkhead And Ring Finger Domains (*CHFR*) gene serves as a predictive biomarker for the efficacy of irinotecan-based systemic chemotherapy in patients with advanced CRC (*n* = 49). The researchers employed a histocultural drug response assay (HDRA) to evaluate the association between *CHFR* promoter methylation and the efficacy of SN38, the active metabolite of irinotecan, in CRC clinical specimens. A significant, positive correlation was observed between *CHFR* relative methylation value (RMV) and tumor inhibition by *SN38* in the HDRA (r = 0.37), with a median inhibition rate of 30.4%. Based on CHFR-RMV, the training cohort was stratified into high and low groups. Tumors in the high CHFR-RMV group showed a significantly greater response to SN38, with a median inhibition rate of 47.0%, compared to 10.4% in the low CHFR-RMV group. *CHFR* promoter methylation was significantly associated with improved clinical response to irinotecan-based chemotherapy, with a disease control rate of 75.0% in the high CHFR-RMV group compared to 42.9% in the low group. Additionally, higher *CHFR* methylation was significantly associated with prolonged progression-free survival (PFS) in patients receiving FLOFIRI plus bevacizumab (2-year PFS: high vs. low group: 50% vs. 0%) [136].

#### 3.1.2. miRNAs

Horak et al. demonstrated that reduced miR-140 expression in tumor tissue was significantly associated with a metastatic phenotype of CRC (*p* = 0.023) and shorter PFS (OR = 0.4) in CRC patients treated with surgery. In their in vitro experiments, upregulation of miR-140 using synthetic mimics in CRC cell lines led to a significant reduction in cell proliferation. Elevated miR-140 levels also significantly enhanced the sensitivity of cancer cells to oxaliplatin and promoted DNA damage accumulation [137]. An interesting study on the effect of miRNA-31 expression on CRC sensitivity to radiotherapy was conducted by Zhang W et al. CRC cell lines (LoVo and HCT116) were irradiated with 0, 2, 4, 6, or 8 Gy, and miRNA-31 expression was assessed after 48 h. In LoVo cells, miRNA-31 levels significantly declined (progressively with increasing radiation doses, reaching 63%, 52%, 43%, and 35%) relative to controls. A similar, significant dose-dependent reduction was observed in HCT116 cells, with expression levels decreasing to 63%, 62%, 43%, and 35%. They further demonstrated that miRNA-31 regulates the radiosensitivity of CRC cell lines by inhibiting STK40 through binding to the 3′ untranslated region of SK40. STK40 significantly, negatively regulated the radiosensitivity of CRC cells. Thus, miRNA-31 and STK40 can be expected to become potential biomarkers of radiotherapy sensitivity [138]. A specific panel of exosomal miRNAs detected in plasma, including miR-100, miR-92a, miR-16, miR-30e, miR-144-5p, and let-7, has demonstrated strong potential for distinguishing oxaliplatin-resistant patients from those who are treatment-sensitive. The study included 210 patients with advanced stage (III–IV) CRC. The combined miRNA panel demonstrated strong discriminatory power, with an AUC of 0.82 (95% CI: 0.75–0.90). These findings indicate that the six identified miRNAs can more accurately distinguish between chemoresistant and chemosensitive CRC patients compared to conventional tumor markers (the accuracy of CEA and CA19-9 was 0.54 and 0.69, respectively) [139]. In a study by Hao YJ et al., the expression of miR-21 in plasma/exosomes was examined to assess its value in predicting CRC recurrence. The recurrence rate and odds ratio (OR) were assessed in 113 patients at all stages of the disease. Compared to patients grouped according to low-level biomarkers, patients with high levels of miR-21 showed significantly higher recurrence rates, such as 18.9% (vs 1.3%) in the high exo-miR-21 group and 37% (vs 1.1%) in the high plasma miR-21 group. The OR were 17.5 and 54.3, respectively [140]. Another study focusing on the evaluation of miR-4442 as a predictive biomarker for CRC recurrence was conducted by Shibamoto J et al. The study enrolled 108 CRC patients before surgical treatment. Plasma levels of miR-4442 were significantly elevated in patients with recurrent CRC and remained high after curative surgery. 3-year RFS in patients with high miR-4442 levels was significantly worse (achieved by 26.8% of patients), compared to patients with low levels (26.8% vs. 63.7%; HR = 2.07, 95% CI: 1.17–3.76). Additionally, a significant moderate, negative correlation was observed between preoperative plasma miR-4442 levels and time to recurrence in CRC patients (rho = −0.544). The diagnostic performance of miR-4442 in distinguishing patients with early recurrence from those without recurrence was high, with an AUC of 0.88 [141].

#### 3.1.3. lncRNAs

LncRNAs are other epigenetic factors showing potential as predictive biomarkers in CRC. Fan C et al. examined lncRNA MALAT1 as a potential biomarker of response to oxaliplatin (Ox)-based chemotherapy in CRC. MALAT1 exhibited elevated expression levels in oxaliplatin-resistant CRC tissues. Additionally, the researchers demonstrated in a xenograft tumor model that MALAT1 knockdown significantly enhances oxaliplatin sensitivity through the miR-324-3p/ADAM17 axis) [142]. Another study demonstrated that lncRNA SNHG7 may act as a competitive endogenous RNA (ceRNA) for miR-181a-5p, promoting GATA6 expression and contributing to anlotinib (ATB) resistance in CRC. They checked the expression of SNHG7 in ATB-resistant, HCT116/ATB and LOVO/ATB CRC cell lines, which was significantly higher in both lines than in control lines. To deepen the analysis, SNHG7 expression was silenced in HCT116/ATB and LOVO/ATB cell lines. Compared to the control, the cell viability of cell lines with silenced SNHG7 (si-SNHG7) was significantly reduced (after 72 h for both lines). The colony formation assay showed that si-SNHG7 cell lines formed significantly fewer clones than control lines [143]. The key role of lncRNA GAS6-AS1 in 5-fluororacil (5-FU) resistance in CRC was elucidated by Zhu Z et al. To investigate the relationship between GAS6-AS1 expression and the objective response rate (ORR) to 5-FU-based chemotherapy, treatment outcomes were assessed in 138 metastatic CRC patients out of a cohort of 316. The overall ORR was 29.8%. GAS6-AS1 expression levels were significantly higher in patients with stable disease or progressive disease (SD + PD ≈ 70%) compared to those who achieved a partial response (PR ≈ 30%). High GAS6-AS1 expression was associated with significantly reduced disease-free survival (DFS) across the patient cohort (hazard ratio (HR) = 2.30). ROC curve analysis further demonstrated the prognostic value of GAS6-AS1, with AUCs of 0.78, 0.69, and 0.66 for 1-, 3-, and 5-year survival predictions, respectively. In vitro experiments showed that overexpression of GAS6-AS1 significantly increased the IC50 of 5-FU, indicating increased drug resistance, while its knockdown reduced the IC50 in CRC cells (*p* < 0.01). Functional assays further confirmed that GAS6-AS1 promotes CRC cell proliferation and facilitates the G1/S phase transition of the cell cycle. Conversely, silencing GAS6-AS1 significantly inhibited these processes. In vivo, GAS6-AS1 overexpression enhanced tumor growth despite 5-FU treatment, whereas its knockdown led to significantly reduced tumor volume and weight. In addition, GAS6-AS1 level was positively correlated with the degree of differentiation in the TNM classification [144]. Shen X et al. in their study found a statistically significant correlation between serum lncRNA DANCR expression and CRC progression in the TNM classification. The relative level of DANCR expression for patients with TNM I and II (*n* = 29) was 2.141, while for TNM III (*n* = 11) it was 2.654 (*p* = 0.019, U-value = 101.5). Moreover, serum levels of DANCR significantly declined following surgical resection or chemotherapy, reaching values comparable to those found in healthy individuals. However, in instances of tumor recurrence, DANCR levels significantly rise again. These findings indicate that serum DANCR holds potential as a predictive biomarker in CRC [127].

### 3.2. Epigenetic Factors in CRC Prognosis

#### 3.2.1. DNA Methylation

DNA methylation biomarkers have emerged as a valuable diagnostic and prognostic tool in many cancer types [145]. Li et al. analyzed DNA methylation status in early-stage CRC patients using targeted bisulfite sequencing. Several genes, including troponin I2 (*TNNI2*), paired box 8 (*PAX8*), GTP binding elongation factor (*GUF1*), Krüppel-like factor 4 (*KLF4*), ecotropic viral integration site 2B (*EVI2B*), and centrosomal protein 112 (*CEP112*), were identified as being associated with the prediction of liver metastases [146]. Interestingly, these genes had previously demonstrated predictive potential across multiple cancer types [147,148,149]. Chung et al. aimed to create a novel prognostic panel for outcomes in CRC patients. Based on multivariable OS analysis, prognostic significance of methylation status of LIM homeobox transcription factor 1 alpha (*LMX1A*)*,* SRY-box transcription factor 1 (*SOX1*), zinc finger protein 177 (*ZNF177*) and NK6 homeobox 1 (*NKX6.1*) was identified. Analysis revealed that methylation of *LMX1A* and *NKX6.1* was significantly related to higher risk of death (HR = 3.30, 95% CI: 1.25–8.76 and HR = 6.06, 95% CI: 2.18–16.88, respectively). Interestingly, methylation of *LMX1A* in early-stage (stage I or II) CRC patients was significantly associated with poorer DFS and 5-year survival. Furthermore, the multivariable analysis revealed that the methylation status of a four-gene panel serves as an independent prognostic factor (HR = 3.43, 95% CI: 1.37–8.57). Specifically, individuals with methylation across all four genes compared to those without methylation of any of the four genes exhibited a significantly reduced DFS and 5-year survival. These results suggested that the combination of the methylation statuses of these genes may constitute a novel prognostic marker in CRC patients [150]. Recent study by Ali et al. indicated a prognostic value of *APC* gene methylation in CRC patients of Indian descent. *APC* promoter methylation was associated with better OS in CRC patients (*p* = 0.035), suggesting its potential role as a favorable prognostic marker. However, authors report data in the results section that differ from those presented in the figures and conclusion. Therefore, the findings of this study should be interpreted with caution [151]. Kumar et al. examined the promoter methylation status of the insulin-like growth factor binding protein 3 (*IGFBP3*) gene in patients with stage II and III CRC after curative surgery. Notably, the *IGFBP3* promoter methylation was significantly associated with poor survival in stage II CRC patients. The mean OS for the methylated group was 22.23 months, compared to 49.15 months in the unmethylated group (HR = 6.43, 95% CI: 0.99–41.94). These findings supported the potential of *IGFBP3* promoter methylation as a valuable prognostic marker for poor survival in CRC [152].

Hu et al. performed a meta-analysis and TCGA analysis to evaluate the association between ras association domain family member 1 (*RASSF1A*) promoter methylation and DFS in patients with rectal and colon adenocarcinomas. In colon adenocarcinoma, patients exhibiting *RASSF1A* promoter methylation demonstrated a significantly shorter DFS compared to those without methylation (mean: 17.4 vs. 30.1 months; HR = 2.25, 95% CI 1.27–3.99). Conversely, the relationship between *RASSF1A* promoter methylation and DFS in rectal adenocarcinoma was not statistically significant (mean: 19.7 vs. 17 months; HR = 1.58, 95% CI 0.69–3.59) [153]. The study conducted by Jamai et al. revealed that methylation of the *ERCC1* and *MGMT* genes, as well as combined methylation of both genes, were significantly associated with a reduced 5-year survival and OS. Moreover, *MGMT* methylation and methylation of both genes were significantly related to reduced 3-year survival [135]. Secreted frizzled-related protein 1 (*SFRP1*) gene promoter methylation was also examined in tumor tissue samples. Univariable analysis identified *SFRP1* promoter methylation as an independent prognostic factor (HR = 17.31, 95% CI: 2.02–148.29), significantly associated with decreased OS, particularly in stage II and III CRC patients [154]. However, Liu et al. presented different findings in study from 2019. Firstly, patients with *SFRP1* hypermethylation compared to those with hypomethylation exhibited significantly higher 5-year and 8-year survival rates (0.68 vs. 0.47 and 0.56 vs. 0.26, respectively). For secreted frizzled-related protein 2 (*SFRP2*), hypermethylation was also associated with significantly higher survival rates at 3, 5, and 8 years (0.93 vs. 0.72; 0.90 vs. 0.60; 0.82 vs. 0.45, respectively). Multivariable Cox regression analysis indicated that *SFRP2* hypermethylation independently predicted a favorable clinical outcome (HR = 0.34, 95% CI: 0.16–0.72). Additionally, co-hypermethylation of both *SFRP1* and *SFRP2* was significantly linked to improved clinical prognosis (HR = 0.33, 95% CI: 0.16–0.69) [155].

A summary of the discussed DNA methylation-based biomarkers for CRC prognosis is provided in Table 3.

#### 3.2.2. ncRNAs

##### miRNAs

Aberrant expression profiles of miRNAs have been increasingly recognized as key contributors to the molecular pathogenesis and tumor progression in CRC [156]. Calvo-López et al. analyzed miR-21 expression in stage III CRC patients who were treated with adjuvant therapy. It was proven that high levels of miR-21 expression were significantly correlated with a better DFS (HR = 2.65, 95% CI: 1.07–6.61) and with a tendency to a better OS (HR = 2.42, 95% CI: 0.75–7.82) in a 3- and 5-year follow-up period. In addition, the study evaluated the impact of prognostic groups (low-risk T1-3/N1vs high risk T4 or N2) on DFS and examined their association with miR-21 expression. Mir-21 high expression was significantly associated with the improved DFS among the poor prognosis group (T4 or N2) [157].

In addition, exosomal miR-150 was identified as an indicator of unfavorable prognosis in CRC patients. Based on TNM classification, 34 individuals were categorized as having stage I–II disease, and 30 were classified as stage III–IV. Lower expression levels of exosomal miR-150 were significantly correlated with shortened OS time (33.3 months versus 43.3 months). In multivariable analysis, plasma exosomal miR-150 was identified as an independent prognostic factor (HR = 2.56, 95% CI: 1.07–6.13). Interestingly, it was also shown that CRC patients with liver metastases exhibited significantly lower levels of exosomal miR-150 compared to those without metastatic disease [158].

Sanjabi et al. investigated plasma-circulating miR-183-5pas a potential predictor of lymph node metastasis (LNM). Firstly, when compared to healthy individuals, miR-183-5p expression was significantly elevated in CRC patients. To assess the prognostic potential of miR-183-5p in predicting LNM, its diagnostic performance was analyzed in distinguishing between stage I/II and stage III CRC patients. ROC curve analysis demonstrated 79% sensitivity and 92% specificity at the optimal threshold (AUC = 0.92, 95% CI: 0.77–0.96, *p* < 0.0001) [159].

Yang and Zhang assessed miR-1253 expression in 121 colon cancer tissues and cell lines using quantitative real-time PCR (qRT-PCR). Firstly, patients with low miR-1253 expression exhibited significantly reduced OS compared to those with high expression levels. Moreover, Cox regression analysis indicated decreased miR-1253 levels as an independent variable associated with poor prognosis in these individuals (HR = 2.56, 95% CI: 1.08–6.11) [160].

Reduced expression of miR-654-3p has been associated with poorer prognosis compared to CRC cases with its overexpression. Univariable analysis revealed that OS was significantly associated with miR-654-3p expression (HR = 0.26, 95% CI: 0.08–0.85, *p* = 0.026). However, in the multivariable model it was not statistically significant (HR = 1.77, 95% CI: 0.16–18.27) [161].

Interestingly, Brînzan et al. investigated the prognostic value of miR-92a, miR-143, and miR-145 expression levels. Notably, CRC patients with high miR-92a expression had a median survival of 36 months, which was significantly shorter than that observed in the low-expression group with median survival of 51 months. Furthermore, the 5-year survival was significantly higher in the high miR-143 expression group (48 months) compared to the low expression group (36 months). Similarly, patients with low miR-145 expression had a median survival of 35 months, which was significantly shorter than the 48-month median observed in the high expression group. However, among the miRNAs analyzed, only miR-92a expression showed a significant association with OS in CRC according to univariable analysis (HR = 0.27, 95% CI: 0.13–0.54) [162].

In a recent study, survival analysis demonstrated that those with high tissue miR-767-5p expression had a noticeably higher 5-year cumulative survival rate compared to those with low expression levels. Additionally, multivariable Cox regression confirmed its role as an independent prognostic marker in CRC (HR = 0.33, 95% CI: 1.07–8.43) [163]. However, elevated miR-675-5p expression was significantly associated with worse clinical outcomes in CRC patients. miRNA expression was examined in 218 primary CRC tissues and 90 normal tissue samples. Survival analysis included 203 patients, while 27 patients were excluded from DFS analysis due to the presence of distant metastases. Univariable Cox regression with bootstrap resampling revealed a HR of 3.18 (95% CI: 1.66–6.11, *p* = 0.001) for DFS and 4.28 (95% CI: 2.33–7.85, *p* = 0.001) for OS in patients with high miR-675-5p expression. Importantly, multivariable Cox analysis, adjusted for tumor location, histological grade, TNM stage, and adjuvant treatment, identified miR-675-5p as an independent prognostic factor. Elevated expression was associated with significantly reduced DFS (HR = 3.07, 95% CI: 1.59–5.92) and OS (HR = 3.87; 95% CI: 2.10–7.13) [164].

MiR-9 expression was analyzed in both tissue and serum of CRC patients. Survival analysis demonstrated that reduced miR-9 expression in tumor and serum samples was significantly associated with shorter survival times compared to patients with higher expression levels. Furthermore, multivariable analysis confirmed that low miR-9 expression independently predicted poorer prognosis in CRC, with HR = 2.19 (95% CI: 1.5–3.19) for tumor samples and HR = 5.59 (95% CI: 1.24–25.12) for serum samples. Notably this was the first study to report on the prognostic value of serum miR-9 in CRC by examining its concordance with matched tumor miR-9 expression [165].

Fu et al. evaluated preoperative circulating miR-449a as a potential prognostic biomarker in CRC patients. The median follow-up period was 35.6 months, with a range spanning from 6.8 to 78.5 months across the cohort. Univariable analysis revealed that lower miR-449a level was significantly associated with reduced 5-year survival (67.4% vs. 76.9%). Importantly, multivariable Cox regression analysis identified decreased circulating miR-449a as an independent prognostic factor (HR = 2.56; 95% CI: 1.15–8.63), regardless of established indicators such as clinical TNM stage and pathological differentiation [166].

MiR-31 expression status was assessed in patients with stage IV CRC. Based on a median miR-31 expression of 3.45 in tumor tissue, a cut-off value of 3.5 was used to dichotomize cases into high and low expression groups. High miR-31 expression was associated with a significantly shorter median survival (20 vs. 38 months). Univariable analysis showed that elevated miR-31 expression was significantly associated with poor prognosis (HR = 2.12, 95% CI: 1.05–4.29). However, this association did not reach statistical significance in multivariable analysis [167].

Chen et al. identified plasma miR-96 and miR-99b as significant factors correlated with OS in metastatic CRC (relative risk (RR) = 39.11, 95% CI: 5.37–284.59; RR = 0.08, 095% CI: 0.03–0.21, respectively). Multivariable Cox regression analysis revealed that elevated plasma miR-96 and reduced miR-99b expression function as independent poor prognostic biomarkers in metastatic CRC (RR = 69.25, 95% CI: 5.31–902.75, RR = 0.16, 95% CI: 0.04–0.69). In addition, miR-96/miR-99b ratio was also evaluated. Among all tested parameters, the miR-96/miR-99b ratio exhibited the highest AUC equal 0.93 (95% CI: 0.86–0.97, *p* < 0.001) in ROC analysis, outperforming individual miRNAs and CEA alone [168].

The prognostic value of miR-215 expression was recently also evaluated. High expression of miR-215 in CRC tissues was significantly associated with improved prognosis (HR = 0.59, 95% CI: 0.35–0.91) [169].

##### lncRNAs

Latest research evaluated several lncRNAs as potential prognostic biomarkers in CRC. The study conducted by Dong et al. aimed to investigate the potential role of the recently identified ALDOA-related specific transcript (ARST) in the pathogenesis of CRC. Higher plasma levels of ARST expression have been associated with significantly lower survival rates and increased mortality over a 5-year follow-up period (HR = 2.00) [170].

The study from 2020 assessed plasma nuclear-enriched abundant transcript 1 (NEAT1) expression and its prognostic value in CRC patients. Elevated serum levels of NEAT1 were significantly associated with reduced OS. Furthermore, multivariate analysis identified serum NEAT1 expression as an independent prognostic indicator (HR = 2.73, 95% CI: 1.20–5.86) [171]. It was shown that higher tissue levels of family with sequence similarity 30 member A (FAM30A) were significantly associated with improved OS in CRC patients (HR = 0.32, 95% CI: 0.12–0.88) [172]. A study evaluating tissue expression of lncRNA among CRC patients found that elevated levels of HCC-associated long non-coding RNA (HANR) was significantly correlated with decreased DFS and OS. Multivariate analysis further confirmed that increased HANR expression independently predicted poorer DFS (HR = 2.31, 95% CI: 1.71–3.96) and OS (HR = 2.50, 95% CI: 1.96–4.11). [173]. Survival analysis conducted by Chu et al. demonstrated that elevated expression of the testis-associated highly conserved oncogenic long non-coding RNA (THOR) was significantly associated with reduced DFS and OS in CRC patients. Multivariable survival analysis confirmed that high THOR expression was independently linked to shorter DFS (HR = 3.51, 95% CI: 1.93–6.37) and OS (HR = 2.49, 95% CI: 1.34–4.35) [174]. In addition, low expression of melanoma highly expressed non-coding RNA (MHENCR) was also significantly related to favorable prognosis in CRC patients (HR = 0.40, 95% CI: 0.18–0.91). Interestingly, MHENCR has been implicated in promoting CRC development and progression, potentially through its interaction with miR-532-3p [175].

In addition, patients with high long intergenic non-coding RNA for kinase activation (Linc-A) expression exhibited significantly poorer survival outcomes when compared to those with low expression. The 5-year survival rate was 42.5% in the high Linc-A group versus 62.5% in the low-expression group, with corresponding median survival times of 39 and 60 months, respectively. Furthermore, 10-year follow-up data showed similar significant differences with a median survival of 39 months in the high Linc-A group and 73 months in the low-expression group, with 10-year survival rates of 30% and 60%, respectively. Moreover, expression of Linc-A was indicated as an independent risk factor (HR = 0.44, 95% CI: 0.19–0.99) [176]. In line with previous findings, elevated long intergenic non-protein coding RNA 1094 (LINC01094) expression correlated with significantly reduced PFS and OS. It was also identified as an independent prognostic factor (HR = 4.36, 95% CI: 1.82–10.44). In addition, it was also shown that LINC01094 may promote CRC cell proliferation, migration, and invasion by interacting with miR-1266-5p and thus modulating secretory leukocyte protease inhibitor (SLPI) oncogene expression [177].

In another study, high expression of ASB 16 antisense RNA 1 (ASB16-AS1) in tissue and plasma and lncRNA actin filament associated protein 1-Antisense RNA 1 (AFAP1-AS1) in tissue were significantly correlated with reduced PFS in CRC patients. Moreover univariable survival analysis revealed, that elevated expression of ASB16-AS1 in plasma (HR = 1.10, 95% CI: 1.05–1.15) and tissue (HR = 1.07, 95% CI: 1.04–1.10), as well as AFAP1-AS1 in plasma (HR = 1.07, 95% CI: 1.01–1.13) and in tissue (HR = 1.07, 95% CI: 1.04–1.10), were significantly associated with decreased OS [118].

A summary of the discussed ncRNA-based biomarkers is provided in Table 4.

## 4. Conclusions

A growing body of evidence supports the potential of epigenetic biomarkers, particularly DNA methylation patterns and non-coding RNAs, as valuable tools for the diagnosis, prognosis, and prediction of treatment response in CRC. Notably, combined biomarker panels have demonstrated improved sensitivity and specificity, particularly in early disease detection.

Among DNA methylation-based markers, SDC2, SEPT9, LINE-1, ERCC1, IGFBP3, NKX6.1, LMX1A, and SFRP2 have demonstrated strong diagnostic, predictive or prognostic relevance. Importantly, SFRP2 hypermethylation was associated with a favorable prognosis, while others, such as NKX6.1 and IGFBP3, correlated with poor survival. In the category of miRNAs, miR-211, miR-197, and miR-21 showed diagnostic value, while miR-140, miR-4442, and miR-21 were identified as predictors of chemotherapy resistance or recurrence. Prognostically, miR-675-5p and miR-150 were linked to poor outcomes, while miR-767-5p and miR-215 indicated favorable prognosis. Several lncRNAs have also shown clinical promise. Molecules such as SNHG14, LINC01485, and ASB16-AS1 demonstrated high diagnostic potential, while MALAT1, GAS6-AS1, THOR, and LINC01094 were associated with patient outcomes.

However, the currently available studies are subject to numerous limitations that preclude their immediate clinical application. Key barriers include suboptimal reporting quality, limited external validation, and a lack of methodological standardization. Most biomarkers have been evaluated by single research groups, with few subjected to independent replication. Moreover, many findings remain inconsistent or contradictory across studies. These challenges highlight the urgent need for large-scale, well-designed clinical validation trials and methodologically aligned protocols for biomarker analysis and reporting.

Robust validation efforts and clinical trials are essential to confirm the reliability of these biomarkers and facilitate their translation into practical tools for CRC management.

## Figures and Tables

**Table 1 cancers-17-02632-t001:** Characteristics of studies on the use of methylation of selected genes as biomarkers useful in detecting CRC.

Studied Biomarker(s)	Biomarker(s) Change	Study Material	Diagnostic Performance		Study Group [*n*]	Country/Nationality	Reference
			Sensitivity (SN) Specificity (SP)	AUC			
**Stool-Based**							
*COL4A2*, *COL4A1*, *TLX2*, *ITGA4*	hypermethylation	Stool	SN = 92.5%; SP = 91.6% (*COL4A2*) SN = 88.8%; SP = 88% (*COL4A1*) SN = 88.8%; SP = 96.4% (*TLX2*) SN = 82.5%; SP = 96.4% (*ITGA4*) SN = 91.3%; SP = 97.6% (*COL4A2* and *TLX2*)	0.97 (*COL4A2*) 0.97 (*COL4A1*) 0.96 (*TLX2*) 0.95 (*ITGA4*) 0.98 (*COL4A2* and *TLX2*)	240 (80 CRC, 77 A, 83 HC) (validation set)	China	[63]
*COL4A2*, *TLX2*	hypermethylation	Stool	SN = 91.3%; SP = 97.6%	0.98	163 (80 CRC, 83 HC)	China	[63]
*SDC2*, *SEPT9*, *VIM*	hypermethylation	Stool	SN = 91.4%; SP = 100%	0.99	181 (83 CRC, 98 HC)	China	[64]
cg13096260 (*SDC2* promoter region) cg12993163 (*SHOX2* gene body region) combined	hypermethylation	Stool	SN = 92.6%; SP = 90% (CRC vs. HC) SN = 93.6%; SP = 90% (CRC (I–II) vs. HC) SN = 66.7%; SP = 90% (AA vs. HC)	0.94 (CRC vs. HC) 0.97 (CRC (I–II) vs. HC) 0.87 (AA vs. HC)	109 (54 CRC, 15 AA, 40 HC) in the validation set	China	[65]
*SEPT9*, *SDC2* (ColoDefense)	hypermethylation	Stool	SN = 92.3%; SP = 93.2%	0.98	92 (39 CRC, 59 HC)—validation set	China	[66]
*SEPT9*, *SDC2*, *SFRP2*	methylation status analysis	Stool	SN = 94.1%; SP = 89.2%	0.94	1142 (180 CRC, 60 AA, 902 HC)	China	[67]
*SDC2*	hypermethylation	Stool	SN = 83.8% (CRC), 87% (CRC I–II); SP = 98%	0.95 (CRC detection)	1110 (359 CRC, 38 AA, 201 NACN, 512 HC)	China	[68]
*PRDM12*, *FOXE1*, *SDC2*	hypermethylation	Stool	SN = 92.8%; SP = 97.2% (CRC vs. HC); SN = 91.9%; SP = 95.2% (CRC, ADD vs. HC)	0.95 (CRC vs. HC) 0.95(CRC, ADD vs. HC)	800 (537 HC, 67 non-ADD, 10 ADD, 138 CRC, 47 IFD)	China	[69]
*SDC2*	hypermethylation	Stool	SN = 90.2% (CRC vs. HC) SN = 89.1% (CRC 0–II vs. HC); SP = 90.2%	0.90 (CRC vs. HC)	585 (245 CRC, 44 P, 245 HC)	South Korea	[70]
*KCNQ5*, *C9orf50*	hypermethylation	Stool	SN = 88.4%; SP = 89.4% (both) SN = 77.3%; SP = 91.5% (*KCNQ5*) SN = 85.9%; SP = 95% (*c9ORF50*)	0.89 (both) 0.85 (*KCNQ5*) 0.90 (*C9orf50*)	460 (20 AA, 198 CRC, 141 HC, 101 SP)	China	[71]
*NDRG4*, *SDC2*	hypermethylation	Stool	SN = 85.5%; SP = 84.6%	0.85	378 (138 CRC, 27 AA, 35 P, 150 OID, 28 HC)	China	[72]
*VIM*	hypermethylation	Stool	SN = 60%; SP = 100%	N/a	79 (49 CRC, 30 HC)	Iran	[73]
*SDC2*, *ADHFE1*, *PPP2R5C*	methylation status analysis	Stool	SN = 91.5%; SP = 90.3%	N/a	274 (47 CRC, 17 AA, 49 A, 161 HC)	Thailand	[74]
*SDC2*, *SFRP2*, *KRAS* (mutation), hemoglobin	hypermethylation (*SDC2*, *SFRP2*)	Stool	SN = 91.4%; SP = 86.1%	N/a	233 (105 HC, 102 CRC, 20 CA, 6 HP)	China	[75]
*SDC2*, *TFPI2*	hypermethylation	Stool	SN = 93.4%; SP = 94.3%	N/a	114 (61 CRC, 53 HC)	China	[76]
**Blood-Based**							
*R16*, *F9*, *F8*, *R13*, *QKI*, *NDRG4*	hypermethylation	Plasma	SN = 67.3%; SP = 98.2%	N/a	263 (114 CRC, 47 AA, 45 BP, 57 HC)	China	[69]
*KCNQ5*, *C9orf50*, *CLIP4*, *ELMO1*, *ZNF582*, *TFPI2*	methylation status analysis	Plasma	SN = 91.7%; SP = 86.7%	0.96	197 (82 GIC, 75 HC, 11 BT, 29 P)	China	[77]
*TFP2019I2*, *NDRG4*	hypermethylation	Peripheral blood	SN = 88%; SP = 92% (*TFPI2*) SN = 86%; SP = 92% (*NDRG4*)	0.94 (*TFPI2*) 0.95 (*NDRG4*)	100 (50 CRC, 50 HC)	Iran	[78]
CpG cg10673833	methylation status analysis	Plasma	SN = 89.7%; 86.8%	0.90	1493 (1021 HC, 29 CRC, 78 APL, 114 NAA, 250 BL)	China	[79]
*C9orf50*, *KCNJ12*, *ZNF132*, *TWIST 1*	methylation status analysis	Plasma	SN = 80%; SP = 97.1% (validation set)	0.91	67 (32 HC, 35 CRC) (validation set)	China	[80]
*MYO1-G*	hypermethylation	Blood	SN = 84.3%, SP = 94.5% (CRC vs. HC)	0.94	673 (272 CRC, 402 HC)	China	[81]
*GALNT9*, *UPF3A*	hypermethylation (*GALNT9*), hypomethylation (*UPF3A*)	Serum	SN = 78.8%; SP = 100% (CRC, AA vs. HC)	0.90	105 (15 NCF, 13 BEN, 11 NAA, 23 D-AA, 19 P-AA, 24 CRC)	Spain	[82]
*HAND1*, *SEPT9*	hypermethylation	Plasma	SN = 93.3%; SP = 80% (*HAND1*) SN = 66.7%; SP = 86.7% (*SEPT9*)	0.85 (*HAND1*) 0.74 (*SEPT9*)	45 (30 CRC, 15 HC)	Iran	[83]
*SEPT9*, *BMP3*	methylation status analysis	Plasma	SN = 80%; SP = 81%	0.85	262 (38 CRC, 46 AA, 119 NAA, 3 SSL, 13 HP, 43 HC)	Brazil	[84]
*SFRP2* (Methyllight)	hypermethylation	Serum	SN = 69.4%, 87.3%	0.82	117 (62 CRC, 55 HC)	China	[85]
*IRF4*, *IKZF1*, *BCAT1*	hypermethylation	Plasma	SN = 73.9%; SP = 90.1% (CRC vs. HC)	0.82	1620 (184 CRC, 616 A, 820 HC)	Australian, Denmark, The Netherlands, Russia	[86]
*DAB1*, *PPP2R5C*, *FAM19A5* (*cfDNA*)	hypermethylation	Peripheral blood	SN = 64.2%; SP = 78.4%	0.76	169 (95 CRC, 74 HC)	China	[87]
*C9orf50*, *KCNQ5*, *CLIP4* (Trimeth)	hypermethylation	Plasma	SN = 85%; SP = 99%	N/a	234 (143 CRC, 91 HC)	Denmark	[89]
*FOXF1* promotor	hypermethylation	Plasma	SN = 78%; SP = 89.5%	N/a	100 (50 CRC, 50 HC)	Iran	[90]

Abbreviations: A-adenoma; AA-Advanced adenoma; ADD-Advanced adenoma; BEN-Bening pathology; BP-Benign polyps; BT-benign tumors; CRC-Colorectal Cancer; GIC-Gastrointestinal cancers; HC-Healthy Controls; IFD-Interfering disease; N/a-not available; NCF-no colorectal findings; OID-Other intestinal diseases; P-Polyps; PP-Perioperative patients; SP-Small polyps.

**Table 2 cancers-17-02632-t002:** Characteristics of studies on the use of ncRNAs as biomarkers useful in detecting CRC.

Studied Biomarker(s)	Biomarker(s) Change	Study Material	Diagnostic Performance		Study Group	Country/Nationality	Reference
			Sensitivity (SN) Specificity (SP)	AUC			
**miRNA**							
miR-129, miR-410, miR-211, miR-139, miR-197	Upregulated: miR-410, miR-211, miR-139, miR-197 Downregulated: miR-1298	Serum	SN = 100% SP = 100% (miR-211); SN = 100%; SP = 100%(miR-197) SN = 70%; SP = 60% (miR-139); SN = 80%, SP = 60% (miR-410) SN = 73%, SP = 73% (miR-129)	1.00 (miR-211), 1.00 (miR-197), 0.73 (miR-139), 0.72 (miR-410) 0.73 (miR-129)	60 (30 CRC, 30 HC)	Iran	[92]
miR-21, miR-210	Upregulated	Serum	SN = 91.4%; SP = 95% (miR-21) SN = 88.6%; SP = 90.1% (miR-210)	0.93 (miR-210) 0.97 (miR-21)	187 (35 CRC, 51 A, 101 HC)	Egypt	[93]
miR-19b, miR-19a, miR-15b, miR-29a, miR-335, miR-18a	Upregulated	Plasma	SN = 85%; SP = 90%	0.92	297 (100 HC, 101 AA, 96 CRC)	Spain	[94]
miR-144-3p, miR-425-5p, miR-1260b	Upregulated: miR-425-5p Downregulated: miR-144-3p, miR-1260b	Plasma	SN = 93.8%; SP = 91.3%	0.95	115 (48 CRC, 47 HC, 20 NC)	China	[95]
miR-21, miR-26a	Upregulated	Serum	SN = 91.8%; SP = 91.7%	0.95	129 (84 CRC, 45 HC)	Egypt	[96]
miR-21	Upregulated	Serum	SN = 95.8%; SP = 91.7%	0.94	96 (48 CRC, 48 HC)	Egypt	[97]
miR-92a-1	Upregulated	Serum	SN = 81.8%; SP = 95.6%	0.91	216 (148 CRC, 68 HC)	China	[98]
miR-1246, miR-451	Upregulated: miR-1246 Downregulated: miR-451	Serum	SN = 100%; SP = 80% (miR-1246) SN = 73%; SP = 80% (miR-451)	0.92 (miR-1246) 0.76 (miR-451)	67 (37 CRC, 30 HC)	Egypt	[99]
miR-18a, miR-21, miR-92a	Upregulated	Serum	SN = 86%; SP = 90% (combined) SN = 84%; SP = 84% (miR-18a) SN = 84%; SP = 90% (miR-21) SN = 66%; SP = 68% (miR-92-a)	N/a (combined) 0.91 (miR-18a) 0.92 (miR-21) 0.67 (miR-92-a)	100 (50 CRC, 50 HC)	Egypt	[100]
miR-378e	Upregulated	Serum	SN = 89%; SP = 80%	0.93	200 (110 HC, 90 HC)	China	[101]
miR-126, miR-1290, miR-23a, miR-940	Upregulated	Serum	SN = 86.6%; SP = 77.1% (miR-126) SN = 83.3%; SP = 85.7% (miR-1290) SN = 89.9%; SP = 74.3% (miR-23a) SN = 90%; SP = 71.4% (miR-940)	0.94 (miR-126) 0.92 (miR-1290) 0.89 (miR-23a) 0.88 (miR-940)	135 (100 CRC, 35 HC)	China	[102]
miR-627-5p, miR-199a-5p	Upregulated	Serum	SN = 87%; SP = 100% (miR-627-5p) SN = 93%; SP = 70% (miR-199a-5p)	0.97 (miR-627-5p) 0.90 (miR-199a-5p)	150 (60 CRC, 60 AA, 30 HC)	Japan	[103]
miR-223, miR-182	Upregulated	Serum	SN = 97.1%; SP = 96.7% (miR-223) SN = 98%; SP = 96% (miR-182)	0.96 (miR-223) 0.95 (miR-182)	65 (35 CRC, 30 HC)	Egypt	[104]
miR-149-3p, miR-607-5p, miR-1246, miR-4488, miR-677-5p	Upregulated: miR-4488, miR-149-3p, miR-1246 Downregulated: miR-607-5p, miR-6777-5p	Stool	SN = 90%; SP = 88% (CRC vs. HC) SN = 82%; SP = 91% (CRC (I–II) vs. HC) (validation cohort)	0.96 (CRC vs. HC) 0.95 (CRC (I–II) vs. HC) (validation cohort)	221 (141 CRC, 80 HC)	Czech Republic	[105]
miR-1290, miR-320d	Upregulated: miR-1290 Downregulated: miR-320d	Plasma	SN = 76.7%; SP = 90.2% (miR-1290) SN = 88.8%; SP = 71.7% (miR-320d)	0.88 (miR-1290) 0.81 (miR-320d)	160 (80 CRC, 30 HC, 50 A)	China	[106]
miR-203a-3p, miR-145-5p. miR-375-3p, miR-200c-3p	Upregulated: miR-203a-3p, miR-145-3p Downregulated: miR-200c-3p, miR-375-3p	Serum	SN = 81.3%; SP = 73.3%	0.89	160 (80 HC, 80 CRC) (validation set)	China	[107]
miR-30e-3p, miR-146a-5p, miR-148a-3p	Upregulated: miR-30e-3p, miR-146a-5p Downregulated: miR-148a-3p	Serum	SN = 80%; SP = 78.7%	0.88	282 (137 CRC, 145 HC)	China	[108]
miR-592	Upregulated	Serum	SN = 82.8%; SP = 78% (CRC vs. HC) SN = 78.6%; SP = 80% (CRC (I–II) vs. HC)	0.84 (CRC vs. HC) 0.80 (CRC (I–II) vs. HC)	184 (50 HC, 134 CRC) (validation set)	China	[109]
miR-135b-5p	Upregulated	Stool	SN = 96.5%; SP = 74.1%	0.87	106 (77 CRC, 29 HC)	China	[110]
miR-193-5p, miR-210, miR-513a-5p, miR-628-3p	Upregulated	Plasma	SN = 90%; SP = 80% (Japan-training cohort) SN = 82%; SP = 86% (Spain-validation cohort)	0.88 (Japan-training cohort) 0.88 (Spain-validation cohort)	117 training cohort (72 EOCRC, 45 HC) 142 validation cohort (77 EOCR, 65 HC)	Japan, Spain	[111]
miR-92a, miR-211, miR-25	Upregulated	Plasma	SN = 71%; SP = 67% (miR-92a) SN = 71%; SP = 92% (miR-211) SN = 75%; SP = 85% (miR-25)	0.77 (miR-92a) 0.79 (miR-211) 0.81 (miR-25)	84 (44 CRC, 40 HC)	Egypt	[112]
hsa-miR-3940-5p	Downregulated	Serum	SN = 93.5%; SP = 82.4%	N/a	130 (20 HC, 70 CRC, 40 BCRC)	Egypt	[113]
miR-15b, miR-21, miR-31	Upregulated	Serum	SN = 95.1%; SP = 94.4% (CRC vs. HC) SN = 85.2%, SP = 82.1% (CA vs. CRC)	N/a	238 (81 CRC, 67 CA, 90 HC) (validation set)	China	[114]
**LncRNA**							
SNHG14	Upregulated	Serum	SN = 98.5%; SP = 90.8%	0.95	130 (70 CRC, 40 BCRC, 20 HC)	Egypt	[113]
Exosomal LINC02418	Upregulated	Plasma	SN = 95.2%; SP = 66.4%	0.90	250 (125 CRC, 125 HC)	China	[116]
MALAT1, PVT1	Upregulated	Serum	SN = 82%, SP = 88% (MALAT1) SN = 90%; SP = 70% (PVT1)	0.91 (MALAT1) 0.85 (PVT1)	280 (140 CRC, 40 AP, 100 HC)	Egypt	[117]
ASB16-ASI, AFAPI-ASI	Upregulated	Plasma	SN = 91.5%; SP = 88% (CRC vs. HC), SN = 88.2%; SP = 62% (early CRC vs. HC) (LncRNA.ASB16-ASI), SN = 87.2%; SP = 84% (CRC vs. HC), SN = 70.6%; SP = 84% (early CRC vs. HC) (LncRNA.AFAPI-ASI)	0.93 (CRC vs. HC), 0.95 (early CRC vs. HC) (LncRNA.ASB16-ASI), 0.92 (CRC vs. HC), 0.93 (early CRC vs. HC) (LncRNA.AFAPI-ASI)	146 (47 CRC-including 17 patients with early-stage CRC, 49 BL, 50 HC)	Egypt	[118]
LINC01485	Upregulated	Whole blood	SN = 98.3; SP = 84%	0.96	85 (60 CRC, 25 HC)	China	[119]
ZFAS, miRNA-200b	Upregulated: ZFAS Downregulated: miRNA-200b	Serum	SN = 98.3%, SP = 96.4% (miR-200b) SN = 98.3%; SP = 92.9% (ZFAS)	0.95 (miR-200b) 0.95 (ZFAS)	88 (28 HC, 60 CRC)	Egypt	[120]
91H, PVT-1, MEG3	Upregulated	Plasma	SN = 82.8; SP = 78.6%	0.88	114 (56 HC, 58 CRC)	China	[121]
ADAMTS9-AS1	Downregulated	Serum	SN = 71.7%; SP = 91.7% (validation set)	0.83 (validation set)	260 (130 HC, 130 CRC)	China	[122]
NKILA	Downregulated	Serum	SN = 82.9%; SP = 72.9%	0.84	90 (70 CRC, 20 HC)	China	[123]
CCAT1, CCAT2, H19, HOTAIR, HULC, MALAT1, PCAT1, MEG3, PTENP1, TUSC7	Upregulated: CCAT1, CCAT2, H19, HOTAIR, HULC, MALAT1, PCAT1 Downregulated: MEG3, PTENP1, TUSC7	Stool	validation set: SN = 74.9%, SP = 94.1% (CRC vs. HC) SN = 68.2%; SP = 83.7% (CRC(I–II) vs. HC) training set: SN = 78.2%, SP = 94.8% (CRC vs. HC) SN = 67.9%; SP = 83.1% (CRC (I–II) vs. HC)	validation set: 0.85 (CRC vs. HC) 0.82 (CRC (I–II) vs. HC) training set: 0.86 (CRC vs. HC) 0.79 (CRC (I–II) vs. HC)	150 (60 CRC, 60 HC, 30 CP)	Iran	[124]
LINC01836	Upregulated	Serum	SN = 65%; SP = 87%	0.81	222 (171 CRC, 51 BA, 138 HC)	China	[125]
lnc-PDZD8-1:5; NEAT1:11; LINC00910:16	Upregulated	whole blood	SN = 74.5%, SP = 80.5%	0.85	186 (85 HC, 101 CRC)	China	[126]
CCAT1, BBOX1-AS1, LINC00698, FEZF1-AS1, UICLM	Upregulated: FEZF1-AS1, UICLM Downregulated: CCAT1, BBOX1-AS1, LINC00698	Plasma	SN = 85% SP = 93% (UICLM), SN = 68% SP = 67% (BBOX1-AS1) SN = 69% SP = 63% (FEZF1-AS1), SN = 67% SP = 62% (CCAT1) SN = 67% SP = 60% (LINC00698)	0.88 (UICLM), 0.72 (BBOX1-AS1) 0.67 (FEZF1-AS1), 0.73 (CCAT1) 0.70 (LINC00698)	60 (30 CRC, 30 HC)	Iran	[92]
DANCR	Upregulated	Serum	SN = 67.5%, SP = 82.5% (*DANCR*)	0.75	130 (50 CRC, 40 HC, 40 CP)	China	[127]
MEG3	Downregulated	Serum	SN = 66.7%; SP = 87.5%	0.79	174 (126 CRC, 48 HC)	China	[128]
lncRNA-ATB	Upregulated	Plasma	SN = 82%; SP = 75%	0.78	148 (74 KC, 74 CRC)	Iran	[129]
**circRNA**							
circ-PNN	Upregulated	Serum	SN = 89.7%; SP = 69% (validation set)	0.83 (validation set)	116 (58 CRC, 58HC) (validation set)	China	[130]
circ-CCDC66, circ-ABCC1,circ-STIL	Downregulated	Plasma	SN = 64.4%; SP = 85.2%	0.78	106 (45CRC, 61HC)	China	[131]
circPAR1, CEA, CA19-9	Downregulated	Plasma	SN = 87.3%; SP = 76.3%	0.88	200 (112CRC, 28 CP, 60 HC)	China	[132]

Abbreviations: AA-Advanced adenoma; AP-Adenomatous polyps; BA-benign adenomas; BL-benign lesions; CA-colorectal adenoma; CP-colon polyps; CRC-Colorectal Cancer; EOCRC-Early onset Colorectal Cancer; HC-Healthy Controls; NC-noncancerous polyps.

**Table 3 cancers-17-02632-t003:** Characteristics of studies on the use of methylation of selected genes as biomarkers useful in colorectal cancer prognosis.

Studied Biomarker(s)	Biomarker Change	Study Material	Diagnostic Performance	Study Group (*n*)	Race/Nationality	Reference
*TNNI2*, *PAX 8*, *AC011298*, *CEP112*, *GUF1*, *KLF4*, *EVI2B*	hypomethylation: *TNNI2*, *PAX 8*, *AC011298*, *CEP112* hypermethylation: *GUF1*, *KLF4*, *EVI2B*	tissue	liver metastasis prediction	CRC (*n* = 59)	China	[146]
*LMX1A*, *SOX1, ZNF177* and *NKX6.1*	hypermethylation	tissue	5-years OS (↓), DFS (↓)	CRC (*n* = 151)	Taiwan	[150]
*APC*	hypermethylation	tissue	OS (↑)	CRC (*n* = 142)	India	[151]
*IGFBP3*	hypermethylation	tissue	OS (↓)	CRC (*n* = 58)	India	[152]
*RASSF1*	hypermethylation	tissue	DFS (↓)	colon adenocarcinoma (*n* = 240), controls (*n* = 38)	TCGA data base	[153]
*MGMT*, *ERCC1*	hypermethylation	tissue	OS (↓)	CRC (*n* = 111)	Tunisia	[135]
*SFRP1*	hypermethylation	tissue	OS (↓)	CRC (*n* = 54)	India	[154]
*SFRP2*	hypermethylation	tissue	OS (↑)	CRC (*n* = 307)	China	[155]

Abbreviations: CRC—Colorectal Cancer; DFS—disease-free survival; OS—overall survival; ↓—decrease in Assessed Study Endpoints; ↑—increase in Assessed Study Endpoints.

**Table 4 cancers-17-02632-t004:** Characteristics of studies on the use of ncRNA-based biomarkers useful in CRC prognosis.

Studied Biomarker(s)	Biomarker Change	Study Material	Assessed Study Endpoints (Associated Change)	Study Group	Race/Nationality	Reference
**miRNA**						
miR-21	high expression	tissue	DFS (↑)	CRC III stage (*n* = 150)	Spain	[157]
exosomal miR-150	low expression	plasma	OS (↓)	CRC (*n* = 64)	China	[158]
miR-183-5p	high expression	plasma	LNM prediction	CRC (*n* = 33) controls (*n* = 13)	Iran	[159]
miR-1253	low expression	tissue	OS (↓)	CRC (*n* = 121)	China	[160]
miR-654-3p	high expression	tissue	OS (↑)	CRC (*n* = 103) controls (*n* = 103)	China	[161]
miR-92a	high expression	tissue	MST (↓)	CRC (*n* = 82)	Romania	[162]
miR-143, miR-145	low expression	tissue	5-year survival (↓)	CRC (*n* = 82)	Romania	[162]
miR-767-5p	low expression	tissue	5-year survival (↓)	CRC (*n* = 133)	China	[163]
miR-675-5p	high expression	tissue	DFS (↓)	CRC (*n* = 176)	Greece	[164]
miR-675-5p	high expression	tissue	OS (↓)	CRC (*n* = 203)	Greece	[164]
miR-9	low expression	tissue	OS (↓)	CRC (*n* = 357)	South Korea	[165]
miR-9	low expression	plasma	OS (↓)	CRC (*n* = 113)	South Korea	[165]
miR-449a	low expression	plasma	5-year OS (↓)	CRC (*n* = 343) controls (*n* = 162)	China	[166]
miR-31	high expression	tissue	MST	CRC IV stage (*n* = 67)	Japan	[167]
miR-96	high expression	plasma	OS (↓)	CRC (*n* = 90) controls (*n* = 20)	China	[168]
miR-99b	low expression	plasma	OS (↓)	CRC (*n* = 90) controls (*n* = 20)	China	[168]
miR-215	low expression	tissue	5-year survival (↓)	CRC (*n* = 214)	China	[169]
**lncRNA**						
ARST	high expression	plasma	5-year survival (↓)	CRC (*n* = 60) controls (*n* = 60)	China	[170]
FAM30A	low expression	tissue	5-year survival (↓)	CRC (*n* = 107)	China	[172]
NEAT1	high expression	plasma	OS (↓)	CRC (*n* = 135)	China	[171]
HANR	high expression	tissue	OS (↓), DFS (↓)	CRC (*n* = 165)	China	[173]
THOR	high expression	tissue	OS (↓), DFS (↓)	CRC (*n* = 103)	China	[174]
MHENCR	high expression	tissue	OS (↓)	CRC (*n* = 143)	China	[175]
Linc-A	high expression	tissue	5-year and 10-year survival (↓)	colon adenocarcinoma (*n* = 80)	China	[176]
LINC01094	high expression	tissue	OS (↓), PFS (↓)	CRC (*n* = 122)	China	[177]
ASB16-AS1	high expression	plasma and tissue	OS (↓), PFS (↓)	CRC (*n* = 47) controls (*n* = 50)	Egypt	[118]
AFAP1-AS1	high expression	tissue	OS (↓), PFS (↓)	CRC (*n* = 47) controls (*n* = 50)	Egypt	[118]

Abbreviations: CRC—Colorectal Cancer; DFS—disease-free survival; LNM—lymph node metastasis; MST-median survival time; OS—overall survival; PFS—progression-free survival; ↓—decrease in Assessed Study Endpoints; ↑—increase in Assessed Study Endpoints.

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
