# Peer review of "The Role of Epigenetic Biomarkers as Diagnostic, Predictive and Prognostic Factors in Colorectal Cancer"

_cancers, 2025, doi:10.3390/cancers17162632_

Round 1

Reviewer 1 Report

Comments and Suggestions for Authors

The aim of this review paper was to „evaluate original studies and meta-analyses published within the last five years to identify the most promising epigenetic biomarkers across these key clinical applications”.

I have read the paper carefully and it is clear that the authors have put a lot of work into writing it, however, I have a few comments.

(1) A general note throughout the text is to italicize the names of the genes themselves, not their explanations. This does not apply to non-coding RNAs. Please carefully review the entire text starting with the abstract of the paper.

(2) The second general remark is that all abbreviations of genes and proteins must be annotated, especially when they first appear in the text; it is practice to place abbreviations also at the end of the article, preferably alphabetically. This will make it easier to understand the content of the works cited.

(3). I believe that the abstract should be slightly shortened.

(4) The general comment also applies to the editing of the entire work, there are too many disjointed parts, subsections and subsections of subsections. Paragraphs are short and cytwana one work. This makes the work difficult to read and it feels like a lot of content is repeated. Some subsections should be shortened, e.g. lines 211-275.

(5) Another general comment is the need to cite Tables in the text of the paper. There are none cited.

(6) From the multiplicity of markers studied from all aspects in the CRC, the names of the markers should also be included in the conclusions.

The Introduction chapter needs rewriting. The division into small subsections is incomprehensible. Such a subsection should be a whole, e.g., dealing only with CRC (without subsections 1.1., 1..2, etc.), moreover, at the end of the Introduction subsection, the purpose of the current review paper should be written. In a word, I would remove all the minor parts of the Introduction.

The subtitle of the second chapter (Diagnosis of CRC) should be referred to only epigenetic changes important in the diagnosis of CRC. The content of the text between lines 277-287 is general and does not add anything interesting to the paper, you can skip and start with line 289, “To date...” Also, please check all the content quoted in the text, e.g. lines 337-341 seems to repeat the work from lines 322-325.

I would also improve the style of the works cited, which is too detailed (as if copied abstract) and thus tedious for the reader. Do not give the names next to each cited paper, the year of the paper, the purpose of their work and only the results. Please focus on the results of the work and cite the author. In addition, for me, the criteria for describing the results from the cited publications are incomprehensible. Please provide these CRITERIA so that the description in the text correlates with the tables. Please organize it, e.g., in a pattern from the most recent to the oldest (as in the Tables) or by the markers studied. Also, please add after each major chapter some commentary of your own on the cited research results.

The tables also need to be changed. Please remove the names from the works cited, just [71] and usually the citations are the last column of the paper, not the first (at the editor's discretion). Abbreviations under tables are better collected alphabetically, as it is easier to find.

Some individual papers are cited for more than a dozen lines and this is unacceptable, e.g. lines 687-697. A frequently repeated phrase is "Another study by ', "In another study, Jamai D et al.", Hagiwara et al. The sense of the papers is lost with such a number of details given, without any commentary, e.g. in lines 725, 808, , 863, etc. Many details are included in the Tables, so this should not be duplicated in the text.

I recommend that the authors rewrite and improve the style of the paper.

I would recommend supplementing the paper with a figure or another table summarizing the authors' own contribution to the work.

Author Response

Comment 1: "A general note throughout the text is to italicize the names of the genes themselves, not their explanations. This does not apply to non-coding RNAs. Please carefully review the entire text starting with the abstract of the paper."

Response 1: Thank you for your valuable feedback. The suggested changes regarding the italicization of gene names have been carefully implemented throughout the manuscript, starting from the abstract.

Comment 2: "The second general remark is that all abbreviations of genes and proteins must be annotated, especially when they first appear in the text; it is practice to place abbreviations also at the end of the article, preferably alphabetically. This will make it easier to understand the content of the works cited."

Response 2: We have ensured that all gene and protein abbreviations are clearly defined upon their first appearance in the text. Additionally, we have created a comprehensive list of abbreviations, presented in alphabetical order at the end of the manuscript, to enhance readability and facilitate better understanding of the content.

Comment 3: "I believe that the abstract should be slightly shortened."

Response 3: The abstract has been adequately shortened to enhance clarity and conciseness while retaining all essential information.

Comment 4: "The general comment also applies to the editing of the entire work, there are too many disjointed parts, subsections and subsections of subsections. Paragraphs are short and cytwana one work. This makes the work difficult to read and it feels like a lot of content is repeated. Some subsections should be shortened, e.g. lines 211-275."

Response 4: We have made a concerted effort to improve the overall structure of the manuscript by reorganizing and consolidating fragmented sections. Several overly detailed or repetitive subsections, particularly those identified by the Reviewer, have been shortened or integrated to enhance clarity and readability throughout the text. However, we have chosen to retain the subsection numbering, as we believe it facilitates navigation within the manuscript and helps readers more easily locate specific information, especially in a detailed review format.

Comment 5: "Another general comment is the need to cite Tables in the text of the paper. There are none cited."

Response 5: We have implemented the suggested revisions.

Comment 6: "From the multiplicity of markers studied from all aspects in the CRC, the names of the markers should also be included in the conclusions."

Response 6: We have implemented the suggested revisions.

Comment 7: "The Introduction chapter needs rewriting. The division into small subsections is incomprehensible. Such a subsection should be a whole, e.g., dealing only with CRC (without subsections 1.1., 1..2, etc.), moreover, at the end of the Introduction subsection, the purpose of the current review paper should be written. In a word, I would remove all the minor parts of the Introduction."

Response 7: The Introduction has been thoroughly revised to improve clarity and coherence. All minor subsections have been removed to ensure a more cohesive narrative. Furthermore, the aim of the current review has been explicitly stated at the end of the revised Introduction section.

Comment 8: "The subtitle of the second chapter (Diagnosis of CRC) should be referred to only epigenetic changes important in the diagnosis of CRC. The content of the text between lines 277-287 is general and does not add anything interesting to the paper, you can skip and start with line 289, “To date...” Also, please check all the content quoted in the text, e.g. lines 337-341 seems to repeat the work from lines 322-325."

Response 8: We have implemented the suggested revisions.

Comment 9: "I would also improve the style of the works cited, which is too detailed (as if copied abstract) and thus tedious for the reader. Do not give the names next to each cited paper, the year of the paper, the purpose of their work and only the results. Please focus on the results of the work and cite the author. In addition, for me, the criteria for describing the results from the cited publications are incomprehensible. Please provide these CRITERIA so that the description in the text correlates with the tables. Please organize it, e.g., in a pattern from the most recent to the oldest (as in the Tables) or by the markers studied. Also, please add after each major chapter some commentary of your own on the cited research results."

Response 9: As you suggested, we have revised the cited studies throughout the manuscript, aiming to streamline the descriptions and focus primarily on the results, rather than providing detailed background or objectives of each publication. We have also removed excessive repetition of author names and years in the main text to improve readability.

We have revised the structure of the tables so that the order of biomarker presentation in the main text now corresponds directly to the order in the tables. We hope this modification improves the overall readability and alignment of the manuscript. In addition, the tables included at the end of most major chapters are intended to serve as concise summaries, consolidating key findings and facilitating easier comparison across studies. We hope this approach enhances the overall organization of the manuscript and strengthens the correlation between the narrative text and the presented data.

Comment 10: "The tables also need to be changed. Please remove the names from the works cited, just [71] and usually the citations are the last column of the paper, not the first (at the editor's discretion). Abbreviations under tables are better collected alphabetically, as it is easier to find."

Response 10: We have implemented the suggested revisions.

Comment 11: "Some individual papers are cited for more than a dozen lines and this is unacceptable, e.g. lines 687-697. A frequently repeated phrase is "Another study by ', "In another study, Jamai D et al.", Hagiwara et al. The sense of the papers is lost with such a number of details given, without any commentary, e.g. in lines 725, 808, , 863, etc. Many details are included in the Tables, so this should not be duplicated in the text."

Response 11: We have revised the manuscript to focus more selectively on the most relevant findings from the cited studies. Lengthy descriptions of individual papers have been shortened, and repetitive phrasing has been reduced. Furthermore, we removed a portion of the data previously included in the main text that was already presented in tabular form.

Comment 12: "I recommend that the authors rewrite and improve the style of the paper."

Response 12: We have implemented the suggested revisions.

Reviewer 2 Report

Comments and Suggestions for Authors

The review article “The Role of Epigenetic Biomarkers as Diagnostic, Predictive and Prognostic Factors in Colorectal Cancer” primarily focuses on evaluating the importance of epigenetic alterations, including gene methylation patterns and non-coding RNA expression, in colorectal cancer diagnosis and prognosis. Overall, the article is well-written with a good compilation of published works in this field. A thorough edits for grammatical and typographical errors will improve its overall quality. For CRC incidence, the most updated data is available for the year of 2022; hence, it would be better to start the introduction with that instead of citing data for the year 2020.

Author Response

Comment: "Overall, the article is well-written with a good compilation of published works in this field. A thorough edits for grammatical and typographical errors will improve its overall quality. For CRC incidence, the most updated data is available for the year of 2022; hence, it would be better to start the introduction with that instead of citing data for the year 2020."

Response: Thank you for your positive evaluation and thoughtful suggestions. We have revised the manuscript to improve its clarity and have addressed grammatical and typographical errors throughout the text. As you recommended, we have updated the Introduction section and replaced the older epidemiological data with the most recent statistics from 2022 to ensure the background information is up to date.

Reviewer 3 Report

Comments and Suggestions for Authors

Dear authors,

Firstly, congratulations on your laborious review of the work. Well done.

Perhaps the review is too extensive and dense.

Although it seems clear that epigenetic biomarkers, especially methylation and ncRNA, can be a tool for diagnosis and response to treatment in the early stages of the disease.

Some of the paragraphs do not contribute anything to the review and the work could be reduced a little, for example, paragraphs 209 to 242 do not contribute anything.

It seems clear that standardization of the procedures used is necessary.

Perhaps it would be best to select a series of biomarkers that the author considers most useful or easiest to implement in clinical practice. Which ones would you choose?

Author Response

Comment: "Firstly, congratulations on your laborious review of the work. Well done.

Perhaps the review is too extensive and dense.

Although it seems clear that epigenetic biomarkers, especially methylation and ncRNA, can be a tool for diagnosis and response to treatment in the early stages of the disease.

Some of the paragraphs do not contribute anything to the review and the work could be reduced a little, for example, paragraphs 209 to 242 do not contribute anything.

It seems clear that standardization of the procedures used is necessary.

Perhaps it would be best to select a series of biomarkers that the author considers most useful or easiest to implement in clinical practice. Which ones would you choose?"

Response: We sincerely thank you for your valuable feedback and thoughtful comments. In response, we have revised and reorganized the manuscript to improve clarity and reduce excessive density. Particular attention was given to eliminating less relevant or redundant paragraphs, including those mentioned by the Reviewer. Additionally, the Introduction section has been substantially modified to enhance focus and coherence. Moreover, to improve clarity and accessibility of the manuscript, we have included a complete list of abbreviations used throughout the text, arranged in alphabetical order, as recommended by another Reviewer.

While we acknowledge the value of highlighting a selected set of the most clinically useful biomarkers, the primary aim of our review is to provide a comprehensive overview of all currently studied and emerging epigenetic biomarkers reported in the literature. This broader scope allows for the identification of promising candidates that may warrant further clinical investigation, even if they have not yet been fully validated or widely adopted in practice.

Round 2

Reviewer 1 Report

Comments and Suggestions for Authors

The authors responded to my doubts and questions posed by myself following reading of the first version of the paper. I would like to thank the authors for the changes they have made and I appreciate their work. Nevertheless, I have a few comments regarding minor changes that could improve the quality and readability of the work (mainly editorial changes). Throughout subsection 2.1.1. “DNA methylation,” please change the order in which gene names and their abbreviations are explained, first the full name of the gene, then the abbreviation in italics in parentheses (as in the Introduction to the paper). Line 134 - the explanation of the MLH1 gene abbreviation is missing, please add it.Lines 220-222 and 1122, 1028, 1040 - please remove the abbreviations CME, CVL, TME, as they are only used once in the entire text and may be somewhat confusing;Line 256 - please write the SEPT9 gene in italics;Lines 339-340 – please delete the entire lines, as they have no text; line 346 – please write the genes TFPI2 and NDRG4 in italics;line 421 – instead of Table1, it should be Table 1 (with a space). I would also suggest listing the marker genes studied in [148] in one column instead of two in Table 3.  

Author Response

Comment: "The authors responded to my doubts and questions posed by myself following reading of the first version of the paper. I would like to thank the authors for the changes they have made and I appreciate their work. Nevertheless, I have a few comments regarding minor changes that could improve the quality and readability of the work (mainly editorial changes). Throughout subsection 2.1.1. “DNA methylation,” please change the order in which gene names and their abbreviations are explained, first the full name of the gene, then the abbreviation in italics in parentheses (as in the Introduction to the paper). Line 134 - the explanation of the MLH1 gene abbreviation is missing, please add it.Lines 220-222 and 1122, 1028, 1040 - please remove the abbreviations CME, CVL, TME, as they are only used once in the entire text and may be somewhat confusing;Line 256 - please write the SEPT9 gene in italics;Lines 339-340 – please delete the entire lines, as they have no text; line 346 – please write the genes TFPI2 and NDRG4 in italics;line 421 – instead of Table1, it should be Table 1 (with a space). I would also suggest listing the marker genes studied in [148] in one column instead of two in Table 3. "

Response: We would like to sincerely thank you for your thoughtful and constructive comments. We truly appreciate the time and effort you devoted to reviewing our revised manuscript and for acknowledging the improvements we have made in response to your initial feedback. In accordance with your latest suggestions, we have carefully addressed all the points you raised:

  • Throughout subsection 2.1.1 “DNA methylation,” we have revised the order of gene names and their abbreviations,
  • On line 134, we have added the full name of the MLH1 gene,
  • The abbreviations CME, CVL, and TME have been removed from lines 220–222, 1028, 1040, and 1122,
  • On line 256, the gene SEPT9 is now correctly written in italics,
  • Empty lines 339–340 have been deleted,
  • On line 346, both gene names TFPI2 and NDRG4 are now in italics,
  • On line 421, we corrected the formatting to “Table 1” with the appropriate spacing,
  • In Table 3, the marker genes from reference [148] are now listed in a single column instead of two.

We believe that these changes have improved the readability and overall quality of the manuscript. Once again, thank you very much for your valuable feedback and for helping us strengthen our work.